# A CLC-ec1 mutant reveals global conformational change and suggests a unifying mechanism for the CLC Cl−/H+ transport cycle

**Tanmay S Chavan[1†], Ricky C Cheng[1†], Tao Jiang[2], Irimpan I Mathews[3], Richard A Stein[4], Antoine Koehl[1], Hassane S Mchaourab[4], Emad Tajkhorshid[2]\*, Merritt Maduke[1]\***

[1]Department of Molecular & Cellular Physiology, Stanford University School of Medicine, Stanford, United States; [2]NIH Center for Macromolecular Modeling and Bioinformatics, Beckman Institute for Advanced Science and Technology, Department of Biochemistry, Center for Biophysics and Quantitative Biology, University of Illinois at Urbana-Champaign, Urbana, United States; [3]Stanford Synchrotron Radiation Lightsource, Stanford University, Menlo Park, United States; [4]Department of Molecular Physiology and Biophysics, Vanderbilt University, Nashville, United States

**\*For correspondence:**
emad@illinois.edu (ET);
maduke@stanford.edu (MM)

[†]These authors contributed equally to this work

**Competing interests:** The authors declare that no competing interests exist.

**Abstract** Among coupled exchangers, CLCs uniquely catalyze the exchange of oppositely charged ions (Cl− for H+). Transport-cycle models to describe and explain this unusual mechanism have been proposed based on known CLC structures. While the proposed models harmonize with many experimental findings, gaps and inconsistencies in our understanding have remained. One limitation has been that global conformational change – which occurs in all conventional transporter mechanisms – has not been observed in any high-resolution structure. Here, we describe the 2.6 Å structure of a CLC mutant designed to mimic the fully H+-loaded transporter. This structure reveals a global conformational change to improve accessibility for the Cl− substrate from the extracellular side and new conformations for two key glutamate residues. Together with DEER measurements, MD simulations, and functional studies, this new structure provides evidence for a unified model of H+/Cl− transport that reconciles existing data on all CLC-type proteins.

## Introduction

CLC transporter proteins are present in intracellular compartments throughout our bodies – in our hearts, brains, kidneys, livers, muscles, and guts – where they catalyze coupled exchange of chloride (Cl−) for protons (H+) (*Jentsch and Pusch, 2018*). Their physiological importance is underscored by phenotypes observed in knockout animals, including severe neurodegeneration and osteopetrosis (*Sobacchi et al., 2007*; *Stobrawa et al., 2001*; *Hoopes et al., 2005*; *Kasper et al., 2005*), and by their links to human disease, including X-linked mental retardation, epileptic seizures, Dent's disease, and osteopetrosis (*Lloyd et al., 1996*; *Hoopes et al., 2005*; *Veeramah et al., 2013*; *Hu et al., 2016*).

CLC-ec1 is a prokaryotic homolog that has served as a paradigm for the family (*Estévez et al., 2003*; *Lin and Chen, 2003*; *Engh and Maduke, 2005*; *Miller, 2006*; *Matulef and Maduke, 2007*). Its physiological function enables resistance to acidic conditions, such as those found in host stomachs (*Iyer et al., 2003*). Like all CLC proteins, CLC-ec1 is a homodimer in which each subunit contains an independent anion-permeation pathway (*Miller and White, 1984*; *Ludewig et al., 1996*;

**eLife digest** Cells are shielded from harmful molecules and other threats by a thin, flexible layer called the membrane. However, this barrier also prevents chloride, sodium, protons and other ions from moving in or out of the cell. Channels and transporters are two types of membrane proteins that form passageways for these charged particles.

Channels let ions flow freely from one side of the membrane to the other. To do so, these proteins change their three-dimensional shape to open or close as needed. On the other hand, transporters actively pump ions across the membrane to allow the charged particles to accumulate on one side. The shape changes needed for that type of movement are different: the transporters have to open a passageway on one side of the membrane while closing it on the other side, alternating openings to one side or the other.

In general, channels and transporters are not related to each other, but one exception is a group called CLCs proteins. Present in many organisms, this family contains a mixture of channels and transporters. For example, humans have nine CLC proteins: four are channels that allow chloride ions in and out, and five are 'exchange transporters' that make protons and chloride ions cross the membrane in opposite directions. These proteins let one type of charged particle move freely across the membrane, which generates energy that the transporter then uses to actively pump the other ion in the direction needed by the cell. Yet, the exact three-dimensional changes required for CLC transporters and channels to perform their roles are still unknown.

To investigate this question, Chavan, Cheng et al. harnessed a technique called X-ray crystallography, which allows scientists to look at biological molecules at the level of the atom. This was paired with other methods to examine a CLC mutant that adopts the shape of a normal CLC transporter when it is loaded with a proton. The experiments revealed how various elements in the transporter move relative to each other to adopt a structure that allows protons and chloride ions to enter the protein from opposite sides of the membrane, using separate pathways. While obtained on a bacterial CLC, these results can be applied to other CLC channels and transporters (including those in humans), shedding light on how this family transports charged particles across membranes.

From bone diseases to certain types of seizures, many human conditions are associated with poorly functioning CLCs. Understanding the way these structures change their shapes to perform their roles could help to design new therapies for these health problems.

---

*Middleton et al., 1996*; *Dutzler et al., 2002*). Studies of CLC-ec1 revealed the importance of two key glutamate residues – 'Glu$_{ex}$' and 'Glu$_{in}$' (*Figure 1A*) in the transport mechanism. Glu$_{ex}$ is positioned at the extracellular entryway to the Cl⁻-permeation pathway, where it acts both as a 'gate' for the transport of Cl⁻ and as a participant in the transport of H⁺ (*Dutzler et al., 2003*; *Accardi and Miller, 2004*). Glu$_{in}$ is located towards the intracellular side of the protein and away from the Cl⁻-permeation pathway, where it appears to act as a H⁺ transfer site (*Accardi et al., 2005*; *Lim and Miller, 2009*).

In CLC transporter crystal structures, Glu$_{ex}$ has been observed in three different positions relative to the Cl⁻-permeation pathway: 'middle', 'up', and 'down'. The 'middle' conformation is observed in the WT CLC-ec1 structure, where Glu$_{ex}$ occupies the extracellular anion-binding site, 'S$_{ext}$' (*Dutzler et al., 2002*; *Figure 1B*). The 'up' conformation is seen when Glu$_{ex}$ is mutated to Gln, mimicking protonation of Glu$_{ex}$; here, the side chain moves upward and away from the permeation pathway, allowing a Cl⁻ ion to bind at S$_{ext}$ (*Dutzler et al., 2003*; *Figure 1B*). The 'down' conformation is seen in the eukaryotic cmCLC structure, where Glu$_{ex}$ plunges downwards into the central anion-binding site, 'S$_{cen}$' (*Feng et al., 2010*; *Figure 1B*). The intracellular anion-binding site, 'S$_{int}$', is a low-affinity site (*Picollo et al., 2009*) and is not depicted.

The rotation of the Glu$_{ex}$ side chain is the only conformational change that has been detected crystallographically in the CLC transporters. A central question, therefore, is whether and how other protein conformational changes contribute to the CLC transport mechanism. In previous work, we used a spectroscopic approach to evaluate conformational changes in CLC-ec1, and we found that raising [H⁺] (to protonate Glu$_{ex}$) caused conformational change in regions of the protein outside of the permeation pathway, up to ~20 Å away from Glu$_{ex}$ (*Elvington et al., 2009*; *Abraham et al.,*

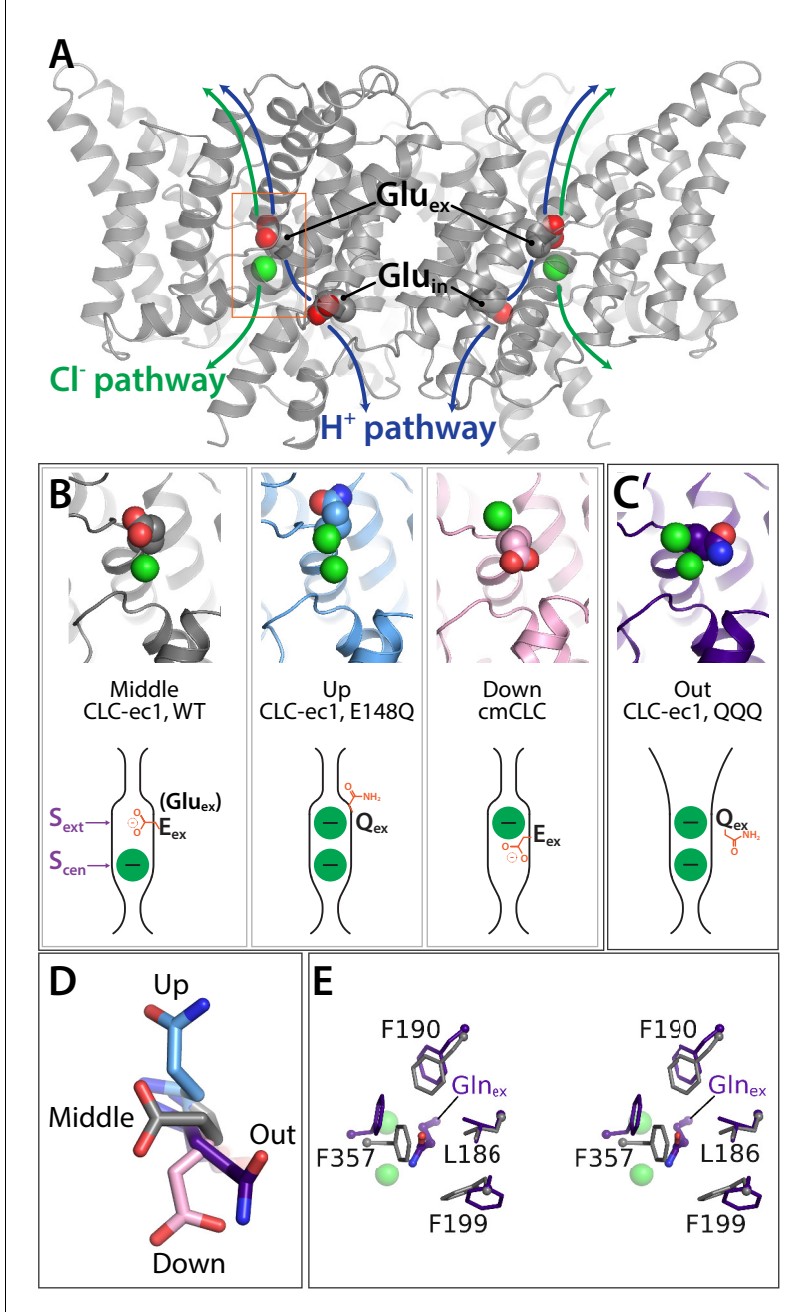

**Figure 1.** Key glutamate residues in CLC transporters. (**A**) CLC-ec1 wild-type structure (PDB 1ots) showing the external and internal glutamate residues (Glu$_{ex}$ and Glu$_{in}$) in the two subunits of the homodimer. Chloride ions are shown as green spheres. Each subunit independently catalyzes Cl$^-$/H$^+$ exchange. The approximate transport pathways for these ions are indicated with green and blue arrows. The orange box frames the close-up view (shown in panel B) of the Cl$^-$-binding sites along with Glu$_{ex}$. (**B**) Glu$_{ex}$ conformations observed in CLC transporters. Three panels showing structures (top panels) and cartoon representations (bottom panels) depicting the three conformations ('middle', 'up', and 'down') adopted by Glu$_{ex}$ in various CLC structures, WT (1ots), E148Q (1otu), and cmCLC (3org). The S$_{ext}$ and S$_{cen}$ anion-binding sites are labeled in the WT cartoon at left. (**C**) QQQ structure reveals a new conformation for Glu$_{ex}$. Structure and cartoon representations as in panel B. (**D**) Overlay of Glu$_{ex}$/ Gln$_{ex}$ conformations seen in QQQ (purple), E148Q (blue), WT (grey) and cmCLC (pink). (**E**) Overlays (stereoview) of WT (grey) and QQQ (purple) illustrate changes in positioning of conserved residues near Glu$_{ex}$.

*2015*). Using a combination of biochemical crosslinking, double electron-electron resonance (DEER) spectroscopy, functional assays, and molecular dynamics (MD) simulations, we concluded that this $H^+$-induced conformational state represents an 'outward-facing open' state, an intermediate in the transport cycle that facilitates anion transport to and from the extracellular side (*Khantwal et al., 2016*).

Here, to obtain a high-resolution structure of the $H^+$-bound conformational state, we crystallized a triple mutant, 'QQQ', in which glutamines replace three glutamates: $Glu_{ex}$, $Glu_{in}$, and E113. E113 is located within hydrogen bonding distance of $Glu_{in}$ and is computationally predicted to be protonated at neutral pH (*Faraldo-Gómez and Roux, 2004*). In contrast to the single-point mutants of $Glu_{in}$ and $Glu_{ex}$, which reveal either no conformational change ($Gln_{in}$) (*Accardi et al., 2005*) or only a simple side-chain rotation ($Gln_{ex}$) (*Dutzler et al., 2003*), the QQQ mutant structure reveals global conformational change, which generates the expected opening of the extracellular permeation pathway. Unexpectedly, this structure additionally reveals new side-chain conformations for both $Gln_{ex}$ and $Gln_{in}$. Based on this new structure, together with MD simulations, DEER spectroscopy, and functional studies, we propose an updated framework for modeling the CLC transport cycle.

## Results

### New conformation of $Glu_{ex}$

The QQQ mutant (E148Q/E203Q/E113Q) was crystallized in the lipidic cubic phase, without any antibody Fab fragment. The structure, determined at 2.6 Å resolution (*Table 1*), reveals an unanticipated change in the conformation of the mutated $Glu_{ex}$ residue, Q148 ($Gln_{ex}$). Instead of occupying the 'up' position, as seen in the structure of the E148Q protein (*Dutzler et al., 2003*), the sidechain has moved away from the permeation pathway and into the hydrophobic core of the protein, a conformation we designate as 'out' (*Figure 1C,D*). This conformation, which has not previously been observed in the CLC transporters, resembles the conformation of $Glu_{ex}$ in the CLC-1 channel structure (*Park and MacKinnon, 2018*). Originally, it was suggested that this 'out' position may be relevant only to CLC channels, due to the steric clashes with conserved residues that the 'out' conformation would generate based on known CLC transporter structures (*Park and MacKinnon, 2018*). However, our new structure reveals that small adjustments in residues 186, 190, 199, and 357 suffice to accommodate $Gln_{ex}$ occupancy in the hydrophobic core (*Figure 1E*).

### Opening of the extracellular vestibule

Analysis of the QQQ structure using HOLE, a program for analyzing the dimensions of pathways through molecular structures (*Smart et al., 1996*), reveals an opening of the extracellular vestibule, increasing accessibility from the extracellular solution to the anion-permeation pathway, in contrast to previously described structures. In the WT protein, two sub-Angstrom bottlenecks occur between $S_{cen}$ and the extracellular side of the protein (*Figure 2A*). In the QQQ protein, these bottlenecks are relieved, widening the pathway to roughly the size of a $Cl^-$ ion (*Figure 2A*, *Video 1*). In contrast, a single point mutation at the $Glu_{ex}$ position (E148Q) relieves only one of the two bottlenecks (*Figure 2A*). This observation is consistent with the QQQ structure representing the CLC-ec1 outward-facing open state.

The extracellular bottleneck to anion permeation is formed in part by Helix N, which together with Helix F forms the anion-selectivity filter (*Dutzler et al., 2002*). Previously, we proposed that generation of the outward-facing open state involves movement of Helix N in conjunction with its neighbor Helix P (at the dimer interface) to widen this bottleneck (*Khantwal et al., 2016*). In addition, Helix N motions have been inferred from experiments on the mammalian antiporter CLC-4 (*Osteen and Mindell, 2008*) and from the gating effects of Helix-N disease causing mutations in CLC-1 (*Wollnik et al., 1997*; *Zhang et al., 2000*; *Pusch, 2002*; *Tang and Chen, 2011*). Disease-causing mutations in Helix N are also found in CLC-Kb, CLC-5, and CLC-7 (*Konrad et al., 2000*; *Leisle et al., 2011*; *Lourdel et al., 2012*). Structural alignment of the QQQ mutant with either E148Q (*Figure 2B–D*) or WT (*Figure 2D*) confirms the movement of these helices. These structural changes involve shifts in highly conserved residues near the anion-permeation pathway, including F190 (Helix G), F199 (Helix H), and F357 (Helix N). The side chains of all repositioned residues show

**Table 1.** Data collection and refinement statistics. Values in parentheses are for highest-resolution shell

| | QQQ |
|---|---|
| **Data collection** | |
| Space group | I222 |
| Cell dimension | |
| a, b, c (Å) | 80.97, 120.44, 122.57 |
| α, β, γ (°) | 90.0, 90.0, 90.0 |
| Resolution (Å) | 28.64–2.62 (2.73–2.62) |
| $R_{merge}$ | 0.164 (0.817) |
| I/ σI | 8.4 (1.6) |
| Completeness % | 97.5 (83.1) |
| Redundancy | 7.4 (4.5) |
| **Refinement** | |
| Resolution (Å) | 39.26–2.62 (2.68–2.62) |
| Number of reflections | 17029 |
| $R_{work}$/$R_{free}$ | 0.192/0.262 |
| | (0.282–0.293) |
| Number of atoms | |
| Protein | 3252 |
| Ions | 3 |
| Water | 55 |
| B-factors | |
| Protein | 64.9 |
| Ions | 83.9 |
| Water | 70.2 |
| r.m.s deviations | |
| Bond length (Å) | 0.005 |
| Bond angle (°) | 1.329 |
| Ramachandran favored | 96.3 |
| Ramachandran outliers | 0 |

good electron density (*Figure 2E*, *Figure 2—figure supplement 1*). Together, these motions widen the extracellular bottleneck (*Figure 2F,G*).

## The intracellular barrier remains constricted

Ion channels have uninterrupted permeation pathways that extend from extracellular to intracellular sides of the membrane, facilitating ion movement down (and only down) the ion's electrochemical gradient. In contrast, to facilitate ion pumping, secondary active transporters must have permeation pathways that are alternately exposed to the extracellular or intracellular sides of the membrane, but never to both simultaneously (*Jardetzky, 1966*; *Tanford, 1983*; *Forrest et al., 2011*). Since Glu$_{ex}$ in the 'out' position has only been observed in a channel structure (CLC-1) before now, it is prudent to question whether this positioning is compatible with an alternating access mechanism. We therefore examined the QQQ structure along the intracellular aspect of the permeation pathway. In WT CLC-ec1, Cl$^-$ permeation to and from the intracellular side is controlled by a constriction formed by conserved residues S107 and Y445, which is thought to act either as a kinetic barrier (*Feng et al., 2010*; *Feng et al., 2012*) or as a gate that opens and closes (*Accardi, 2015*; *Basilio and Accardi, 2015*). In QQQ, this intracellular constriction is unchanged compared to WT CLC-ec1 and is narrower than that observed in the CLC-1 channel (*Figure 3*, *Figure 3—figure*

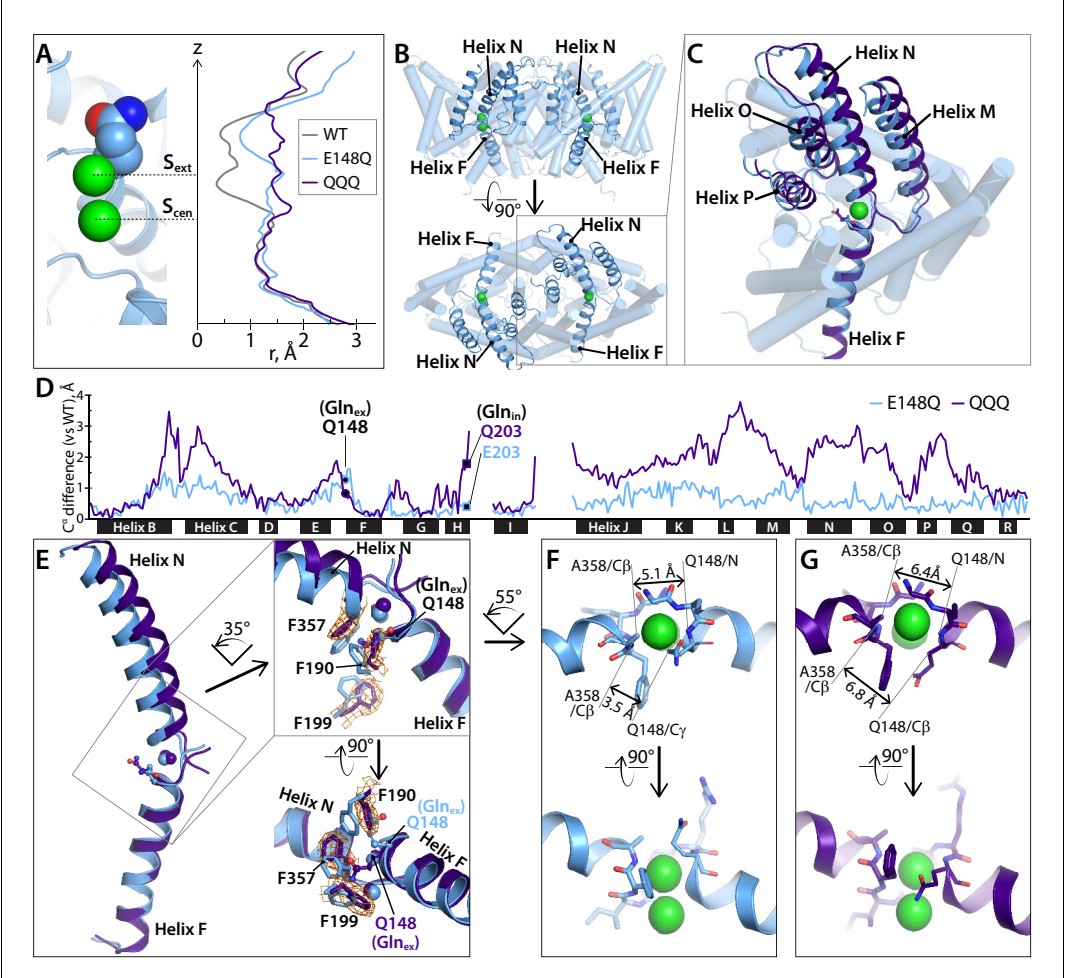

**Figure 2.** Helices M-N-O-P move to widen the extracellular vestibule. (**A**) Comparison of pore radius profiles of CLC-ec1 with $Glu_{ex}$ in the 'middle' (WT), 'up' (E148Q), and 'out' (QQQ) positions. The pore-radius profile, calculated using HOLE, is aligned with the structure at left (E148Q) to indicate the position of the $S_{ext}$ and $S_{cen}$ anion-binding sites. (**B**) Two views of the CLC-ec1 E148Q structure, with ribbons highlighting helices M-N-O-P (helices located in the extracellular half of the protein) and F (located in the intracellular half). (**C**) Overlay of the CLC-ec1 QQQ structure (purple) with E148Q (blue) following structural alignment using segments of helices B, F, G, and I (residues 35–47, 153–164, 174–190, and 215–223). (**D**) Plot of the differences in Cα positions between WT CLC-ec1 and QQQ (purple) or between WT and E148Q (blue), based on the B-F-G-I structural alignment. The black bars indicate 17 of the 18 alpha helices. (Density for Helix A is absent in QQQ.) The locations of $Gln_{ex}$ and $Gln_{in}$ are marked as solid-black circles and squares, respectively. The differences in the H-I and I-J linkers are not shown here but are discussed below and in *Figure 5—figure supplement 2*. (**E**) Zoomed-in view of the structural overlay between CLC-ec1 QQQ and E148Q, showing changes in the positions of conserved residues F190, F199, and F357 and the corresponding electron density from the QQQ structure determination. Additional views are shown in *Figure 2—figure supplement 1*. (**F, G**) Illustration of interatom distances at the extracellular bottleneck that are increased in QQQ (**G**) compared to E148Q (**F**). Comparisons of Cl⁻ coordination distances are shown in *Figure 2—figure supplement 2*.

The online version of this article includes the following figure supplement(s) for figure 2:

**Figure supplement 1.** Comparison of the CLC-ec1 QQQ structure to the E148Q ($Glu_{ex}$ to Gln) structure in the vicinity of the $S_{ext}$ binding site.

**Figure supplement 2.** Cl⁻ binding sites in QQQ CLC-ec1.

*supplement 1*). This maintained intracellular constriction supports QQQ as a viable representative for a transporter intermediate.

## Comparison of QQQ and E148Q Cl⁻ binding and transport rates

The widening of the extracellular Cl⁻ entryway in QQQ is accompanied by subtle changes in the $S_{ext}$ Cl⁻-binding site (*Figure 2—figure supplement 2*). We therefore hypothesized that Cl⁻ binding to this site might be altered. To test this hypothesis, we used isothermal titration calorimetry (ITC) to

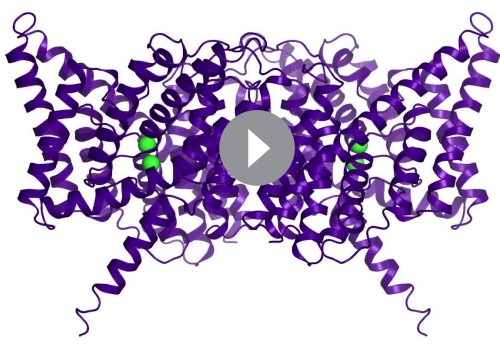

**Video 1.** QQQ structure. $Gln_{ex}$ and inner-gate residues are shown as sticks. The pore profile detected using Caver with probe radius 1.2 Å, starting from $S_{cen}$, is indicated in yellow.

https://elifesciences.org/articles/53479#video1

compare Cl⁻ binding to QQQ and E148Q ($Glu_{ex}$ to $Gln_{ex}$), which both show the $S_{ext}$ site occupied by Cl⁻ (in contrast to WT CLC-ec1, where the $S_{ext}$ site is occupied by $Glu_{ex}$). We found that QQQ and E148Q bind Cl⁻ with similar affinities ($K_d = 138 \pm 26$ μM and $116 \pm 6$ μM, respectively) (*Figure 4A*). Since these measurements do not distinguish between binding to $S_{cen}$ versus binding to $S_{ext}$, we also attempted to distinguish binding at the different sites crystallographically, making use of the anomalous signal obtained from Br⁻ binding, as has been done previously (*Lobet and Dutzler, 2006*). However, we have not been successful in our attempts to produce well-diffracting QQQ crystals in the presence of Br⁻. Regardless, while we cannot make specific conclusions about the $S_{ext}$ site, we can conclude that overall binding of Cl⁻ to QQQ and E148Q CLC-ec1 occurs with similar affinity.

The widening of the extracellular Cl⁻ entryway in QQQ (*Figure 2*) predicts that Cl⁻ transport through QQQ will be faster than through E148Q, if extracellular gate-opening is a rate-limiting step. On the other hand, if Cl⁻ transport through

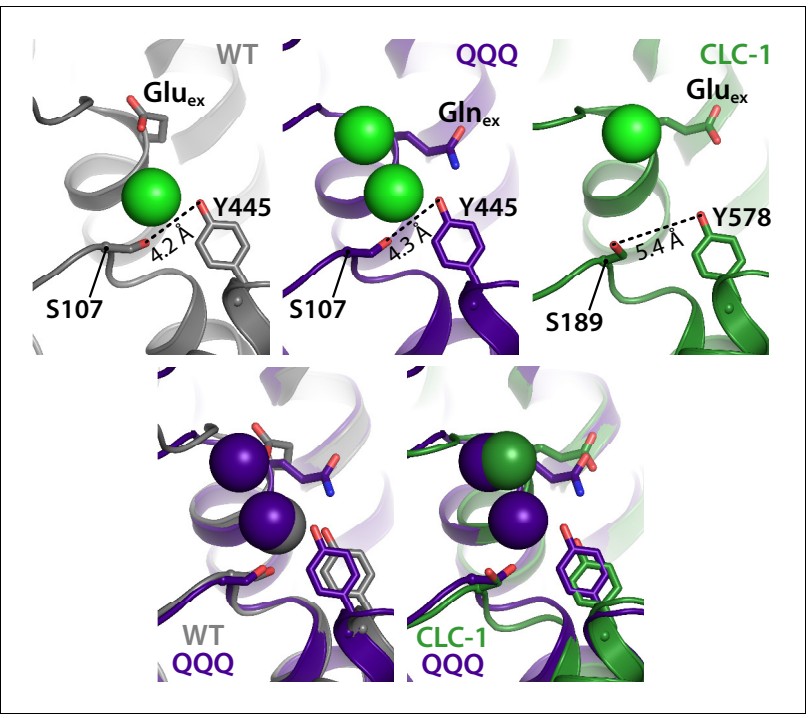

**Figure 3.** The intracellular constriction remains narrow in QQQ. The inter-residue distance between S107 and Y445, conserved residues forming the intracellular constriction, is similar in WT and QQQ (grey and purple in top left panels and overlay below). The equivalent positions in the CLC-1 channel (green, top right panel) are separated by a greater distance, despite the similarity in $Glu_{ex}$ conformation between CLC-1 and QQQ (overlay in bottom right panel).

The online version of this article includes the following figure supplement(s) for figure 3:

**Figure supplement 1.** Pore profiles for CLC-1, QQQ, and WT CLC-ec1.

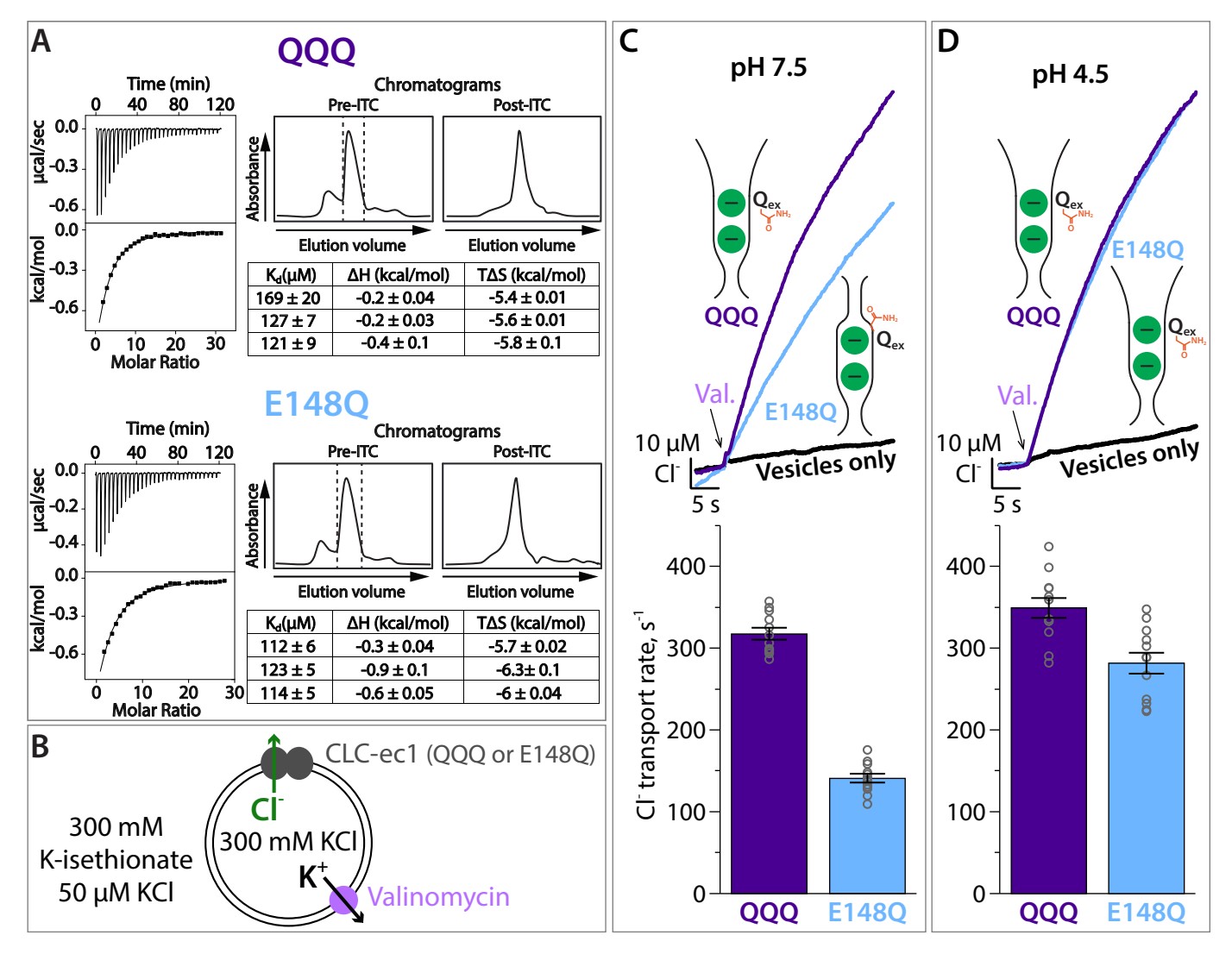

**Figure 4.** Cl⁻ binding and transport by QQQ and E148Q CLC-ec1. (**A**) Cl⁻ binding measured by ITC for QQQ (top panels) and E148Q (bottom panels), at pH 7.5. At left are shown representative data from ITC experiments. At right are shown representative size-exclusion chromatograms of samples before and after the ITC experiment (pre- and post-ITC chromatograms). The dotted line in the pre-ITC chromatogram represents the fraction that was collected for use in the ITC experiment. The sample was run again following the ITC experiment to confirm the sample was stable throughout the experiment. The summary tables show results for 3 experiments performed on 3 separate protein preparations. Averages (± SEM) for E148Q ($K_d = 116 \pm 6$ µM) and QQQ ($K_d, = 138 \pm 26$ µM) are not statistically different (p = 0.19). (**B**) Cartoon depiction of the Cl⁻ flux assay. The Glu$_{ex}$ mutants E148Q and QQQ catalyze downhill transport of Cl⁻. Bulk movement of Cl⁻ is initiated by the addition of valinomycin, and extravesicular Cl⁻ is quantified using a Ag·AgCl electrode. (**C**) At pH 7.5, Cl⁻ transport through the CLC-ec1 QQQ mutant is ~2 fold faster compared to E148Q. The top panel shows representative primary data from the flux assay. The bar graph shows summary data, ± SEM, with n = 12 for each mutant (open circles, samples from 3 independent protein preparations). (**D**) At pH 4.5, favoring protonation of glutamate residues, Cl⁻ transport through E148Q increases by 2-fold, to within 20% the Cl⁻-transport rates by QQQ. The bar graph shows summary data, ± SEM, with n = 12 for each mutant (open circles, samples from 3 independent preparations).

The online version of this article includes the following source data for figure 4:

**Source data 1.** Contains the source data for the transport rates shown in *Figure 4C,D*.

E148Q and QQQ have the same rate-limiting step, then Cl⁻ transport rates, like Cl⁻ binding affinities, should be similar. Experimental measurements revealed that QQQ Cl⁻ transport rates are ~2 fold faster than E148Q transport rates at pH 7.5, the pH at which the binding studies were performed (*Figure 4B,C*; *Figure 4—source data 1*), indicating that the two mutants have different rate-limiting steps. If this difference in transport rates is due to the wider extracellular Cl⁻ entryway in

QQQ compared to E148Q, then lowering the pH – to allow E148Q to adopt the QQQ-like (outward-facing open) conformation – should increase the Cl⁻ transport rate. Consistent with this prediction, transport rates of the E148Q mutant increase by 2-fold at pH 4.5 (*Figure 4D*; *Figure 4—source data 1*). These results support the conclusion that E148Q CLC-ec1 (and by extension WT CLC-ec1) undergoes an opening of the extracellular vestibule at low pH. We note that the relatively slow Cl⁻ transport by both E148Q and QQQ compared to WT CLC-ec1 (~300 $s^{-1}$ versus 2200 $s^{-1}$) is not surprising, given that these $Glu_{ex}$ mutants lack a negatively charged carboxylate to compete Cl⁻ out of the permeation pathway.

## Overall conformational change in QQQ

To evaluate overall conformational changes in the QQQ structure, we generated difference distance matrices, which provide comparisons that are independent of the structural alignment method (*Nishikawa et al., 1972*). Overall, comparison of QQQ to WT CLC-ec1 using the difference distance matrices confirms a hot spot of conformational change at Helices K-N (as illustrated in *Figure 2*) and highlights additional changes at G, H, I, and Q (*Figure 5A*; *Figure 5—figure supplement 1*; *Figure 5—source data 1*). In contrast, comparison of single-Glu mutant structures to WT reveals only minor (≤0.8 Å) changes (*Figure 5B*).

CLCs, like many secondary active transporters, are comprised of pairs of inverted structural repeats (*Duran and Meiler, 2013*; *Forrest, 2015*). These repeats are homologous domains that are inserted with inverted orientation into the membrane, related to one another by an axis of pseudo-symmetry along the membrane plane (*Figure 5C*). Conformational exchange of the repeats, with the first repeat adopting the conformation of the second and vice versa, has been shown in other transporters to convert the overall protein structure from outward- to inward-facing, thus facilitating the alternating access mechanism required to achieve secondary active transport (*Forrest et al., 2008*; *Crisman et al., 2009*; *Forrest and Rudnick, 2009*; *Palmieri and Pierri, 2010*; *Radestock and Forrest, 2011*). Since all previous CLC transporter structures appear 'occluded' (neither inward- nor outward-facing), it is uncertain a priori whether conformational swap will occur upon transition to the outward-facing state in the CLCs. Within each repeat of the QQQ structure, significant changes occur compared to the equivalent repeat in WT CLC-ec1, particularly in Helices H and Q (*Figure 5D*). However, the repeats have not interconverted (*Figure 5—source data 2*). In addition to the changes within each repeat domain, there are substantial changes between the two repeats (relative to one another), most strikingly between Helices G-I of Repeat 1 and Helices J-N of Repeat 2 (*Figure 5D*).

Within Repeat 2, the major change occurs at Helices O-Q. This region is of mechanistic interest because a cross-link between residue 399 on Helix O and 432 on Helix Q is known to inhibit transport through coupling to the inner gate (*Basilio et al., 2014*). Interestingly, although we find that Helices O and Q move with respect to the rest of Repeat 2, residues 399 and 432 do not change position with respect to one another and do not couple to movement at the inner gate (*Figure 5—figure supplement 3A–D*). Thus, our data suggest that the Helix-O motions involved in transition to the outward-facing state are distinct from those postulated to facilitate inner-gate opening. While the mechanistic details linking Helix O-Q movements between outward- and inward-facing states are currently unknown, it is of interest to note that many disease-causing mutations occur in this segment, in both CLC channels (*Saviane et al., 1999*; *Pusch, 2002*; *Bignon et al., 2020*) and transporters (*Lourdel et al., 2012*; *Veeramah et al., 2013*). In Helix O, for example, mutation of a highly conserved glycine residue occurring mid-helix can cause Dent's disease (CLC-5, *Smith et al., 2009*) or Bartter syndrome (CLC -Kb, *Lin et al., 2009*). In WT CLC-ec1, the helix is kinked at this glycine; in QQQ, the helix is straight (*Figure 5—figure supplement 3E,F*).

Within Repeat 1, the major change occurs at Helices G-I. This segment also changes position relative to Repeat 2 (*Figure 5D*). Helix H is of mechanistic interest because it contains the $Glu_{in}$ residue that is thought to transfer H⁺ from the intracellular solution to $Glu_{ex}$ (*Accardi et al., 2005*; *Accardi, 2015*) and because Helix H is one of the most highly conserved regions of the protein (*Dutzler et al., 2002*). The movement of Helices G-I relative to Repeat two is illustrated with structural overlays in *Figure 6A,B*. This outward movement provides space for $Gln_{ex}$ to move to the 'out' position (*Figure 6B*). The movement of Helices G-I relative to other helices in Repeat one is illustrated in *Figure 6C,D*. This movement releases the interaction between residue 113 (Helix D) and $Gln_{in}$ (Helix H) (*Figure 6E*). Strikingly, $Gln_{in}$ moves into the hydrophobic core of the protein, to within

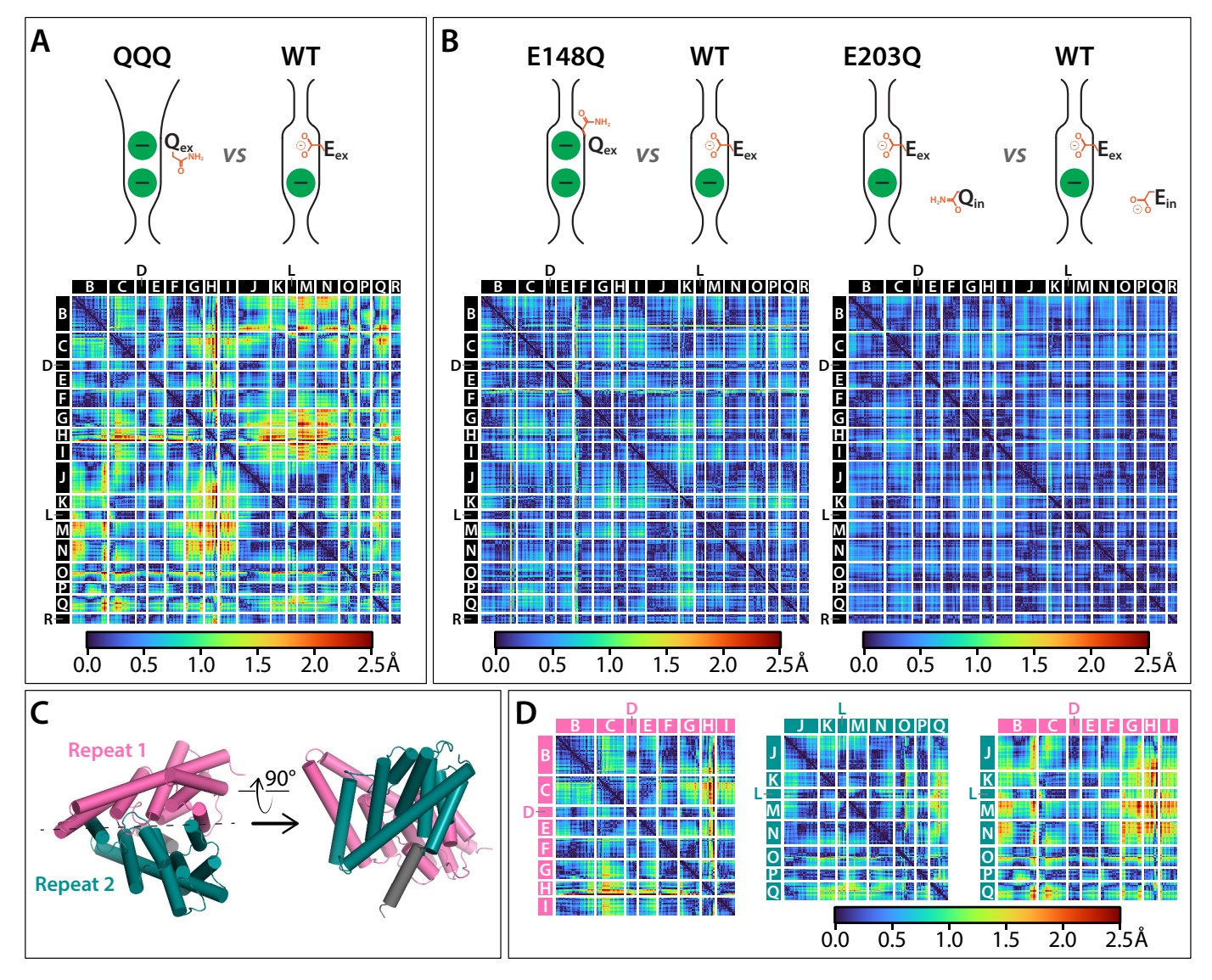

**Figure 5.** Cα difference distances matrices. Color-blind friendly versions of these plots are shown in *Figure 5—figure supplement 1*. Difference distance matrix comparing WT CLC-ec1 (1ots) to QQQ reveals intramolecular rearrangements throughout the protein. Differences are between Cα residues on helices; differences in linker regions are discussed in *Figure 5—figure supplement 2*. (B) Comparison of WT to E148Q (1otu) or E203Q (2fec) reveals only minor (≤0.8 Å) changes. (C) Illustration of inverted topology repeat domains in CLC-ec1. Repeat 1 (Helices B-I, pink) is arranged pseudosymmetrically with Repeat 2 (teal). Helix R (grey) is not part of the repeat domains. Helix A, the N-terminal cytoplasmic helix, is also not part of the repeats; it is not resolved in the QQQ structure and not shown here. (D) Changes occur both within and between the inverted topology repeat domains. Matrices are the same as in panel A, laid out to focus on comparing changes within each Repeat domain (left and middle panels) or between the Repeat domains (right panel). Helices G-I (Repeat 1) and J-N (Repeat 2) move substantially relative to one another, while the equivalent sets (O-Q in repeat 2 and B-F in repeat 1) undergo more modest relative changes.

The online version of this article includes the following source data and figure supplement(s) for figure 5:

**Source data 1.** Excel spreadsheet of difference distance matrices shown in *Figure 5*.

**Source data 2.** Summary of RMSD values between inverted Repeat domains in WT and QQQ.

**Figure supplement 1.** Cα difference distances matrices with color-blind friendly color palette.

**Figure supplement 2.** Linker rearrangements in the QQQ structure are observed at the extracellular I-J linker and the intracellular H-I linker.

**Figure supplement 3.** Conformational change at Helices O-Q.

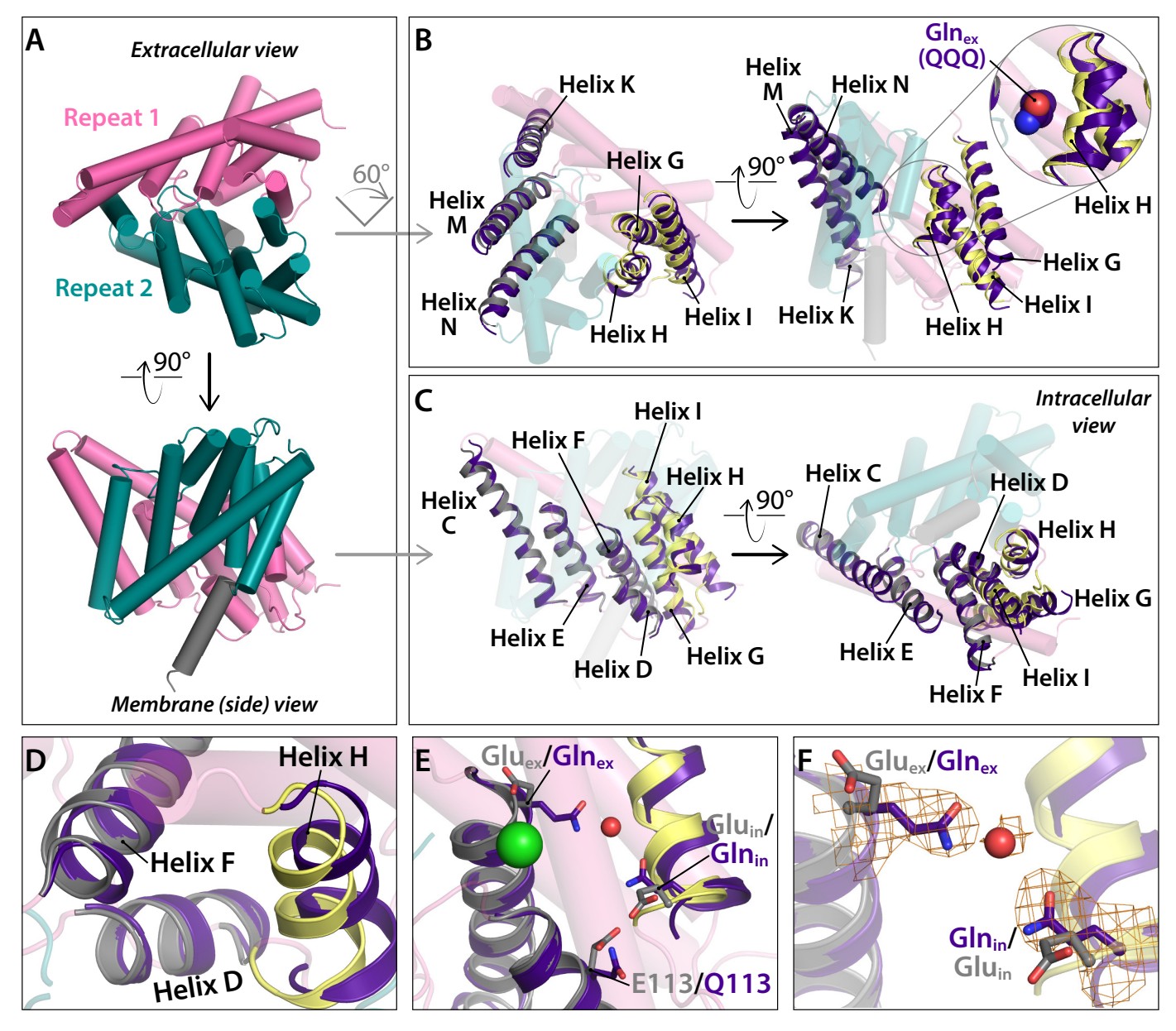

**Figure 6.** Conformational change at Helices G-I and repositioning of Gln$_{in}$ (Helix H). (**A**) View of one CLC subunit, highlighting Repeat 1 in pink and Repeat 2 in teal. Helix R (not part of the repeats) is shown in grey. (**B**) Movement of Helices G-I (part of Repeat 1) relative to Helices K, M, N (Repeat 2). The compared helices are shown in ribbon, with QQQ in purple and WT in gray (Helices K, M, N) or yellow (G–I). Other helices (WT) are shown as transparent cylinders. The inset illustrates how the movement of Helix H away from Helix N creates space to avoid steric conflict with Gln$_{ex}$ in the 'out' position. (**C**) Movement of Helices G-I relative to other helices in Repeat 1. The compared helices are shown in ribbon, with QQQ in purple and WT in gray (Helices C-E) or yellow (G–I). Other helices (WT) are shown as transparent cylinders. (**D**) Close-up view showing movement of Helix H (containing Gln$_{in}$) away from Helix D. (**E**) Conformational change at Gln$_{in}$ moves it away from Q113. (**F**) Movement of Gln$_{in}$ to the hydrophobic core of the protein brings it to within 6 Å of Gln$_{ex}$. Electron density for Gln$_{ex}$, Gln$_{in}$, and an intervening water molecule is shown in mesh.

The online version of this article includes the following figure supplement(s) for figure 6:

**Figure supplement 1.** Crystallographic water molecules near Gln$_{in}$.

6 Å of $Gln_{ex}$ (*Figure 6F*). This change is accompanied by a rearrangement of water molecules in the internal core (*Figure 6—figure supplement 1*).

## Validation of QQQ conformational change in WT CLC-ec1 using DEER spectroscopy

Our working hypothesis is that the QQQ mutant structure mimics the outward-facing open intermediate in the WT CLC transport cycle. When working with a mutant, however, one always wonders whether any conformational change observed is relevant to the WT protein. We therefore used DEER spectroscopy to evaluate conformational change in WT CLC-ec1. DEER spectroscopy is advantageous because it can evaluate conformational change by site-directed spin labeling, without the constraints of crystallization. Accurate distance distributions can be obtained for spin labels separated by ~20–70 Å (*Jeschke, 2012*; *Mishra et al., 2014*; *Stein et al., 2015*). Since CLC-ec1 is a homodimer ~100 Å in diameter, a simple labeling strategy with one spin label per subunit can provide a sample with optimally spaced probes for distance-change measurements. For example, the extracellular sides of Helices N and O (*Figure 2B,C*) are separated by ~50 and 35 Å respectively from their correlates in the other subunit. To test the hypothesis that these helices move, we generated WT CLC-ec1 (cysteine-less background) with spin labels at positions 373, 374 (Helix N) and 385 (Helix O) and performed DEER measurements under two conditions, pH 7.5 and pH 4.5. The rationale for this experimental strategy is that pH 4.5 will promote protonation of $Glu_{ex}$ and $Glu_{in}$, thus favoring a global conformation comparable to that observed in the QQQ structure (*Figure 7A*).

Consistent with our hypothesis, spin labels on Helices N and O exhibited pH-dependent changes in distance distributions in the direction predicted by the QQQ structure (*Figure 7B–D*). To provide a more direct comparison of WT to QQQ, we additionally made measurements on spin-labeled QQQ samples (cysteine-less background). At all three positions, the DEER distance distributions showed little to no pH dependence, and they resembled the distributions observed with WT at pH 4.5 (*Figure 7B–D*). Similar results were obtained for a spin label on Helix P (*Figures 2C* and *7E*), which had been shown by cross-linking experiments to move during the CLC-ec1 transport cycle (*Khantwal et al., 2016*), thus providing further support for the relevance of the conformational changes observed in QQQ. Finally, to test the predicted conformational change at the intracellular side near the $H^+$ permeation pathway (Helix G-I movements, *Figure 6C*), we examined a spin label on Helix G. (Helix G inter-subunit distances are better suited to DEER measurements than Helix H inter-subunit distances.) Once again, the WT protein showed a pH-dependent shift in the direction predicted by the QQQ structure, and the QQQ protein showed distance distributions resembling those of WT at pH 4.5 (*Figure 7F*). Taken together, the DEER distance distributions provide strong support for the conclusion that the QQQ structure represents a WT CLC-ec1 conformation.

## Analysis of water connections to $Gln_{ex}$

Previous computational studies indicated that water wires can transiently bridge the $Glu_{ex}$ and $Glu_{in}$ residues separated by 12.8 Å in the CLC-ec1 WT structure, which may serve as the pathway for $H^+$ transfer (*Wang and Voth, 2009*; *Han et al., 2014*; *Jiang et al., 2016*). The proximity of these residues in the QQQ structure (*Figure 6F*, *Figure 6—figure supplement 1*) motivated us to re-evaluate this phenomenon. In our previous studies on WT CLC-ec1, extended MD simulations revealed that water spontaneously enters the hydrophobic core of the protein and transiently and repeatedly forms water wires connecting $Glu_{ex}$ and $Glu_{in}$ (*Han et al., 2014*; *Jiang et al., 2016*). Analogous simulation of the QQQ mutant revealed a dramatic and unanticipated result: water penetration into the hydrophobic core of the protein is greatly increased, and water pathways directly connect bulk water in the intracellular solution to $Gln_{ex}$, without requiring intermediate connection to $Gln_{in}$ (*Figure 8A–C*). These water pathways were observed frequently during our 600-ns simulation (*Figure 8D*, *Video 2*, *Figure 8—figure supplement 1—source data 1*); in contrast, such water pathways were not observed in our previous 400-ns WT simulations (*Han et al., 2014*; *Jiang et al., 2016*). The number of water molecules needed to reach bulk water follows a normal distribution, with chains of 5 or six water molecules predominating (*Figure 8D*). In contrast, the majority of water wires connecting $Glu_{ex}$ to $Glu_{in}$ in the WT simulation involved seven or more water molecules (*Han et al., 2014*; *Jiang et al., 2016*). Moreover, the occurrence of water pathways in the QQQ simulation (36.5%) is

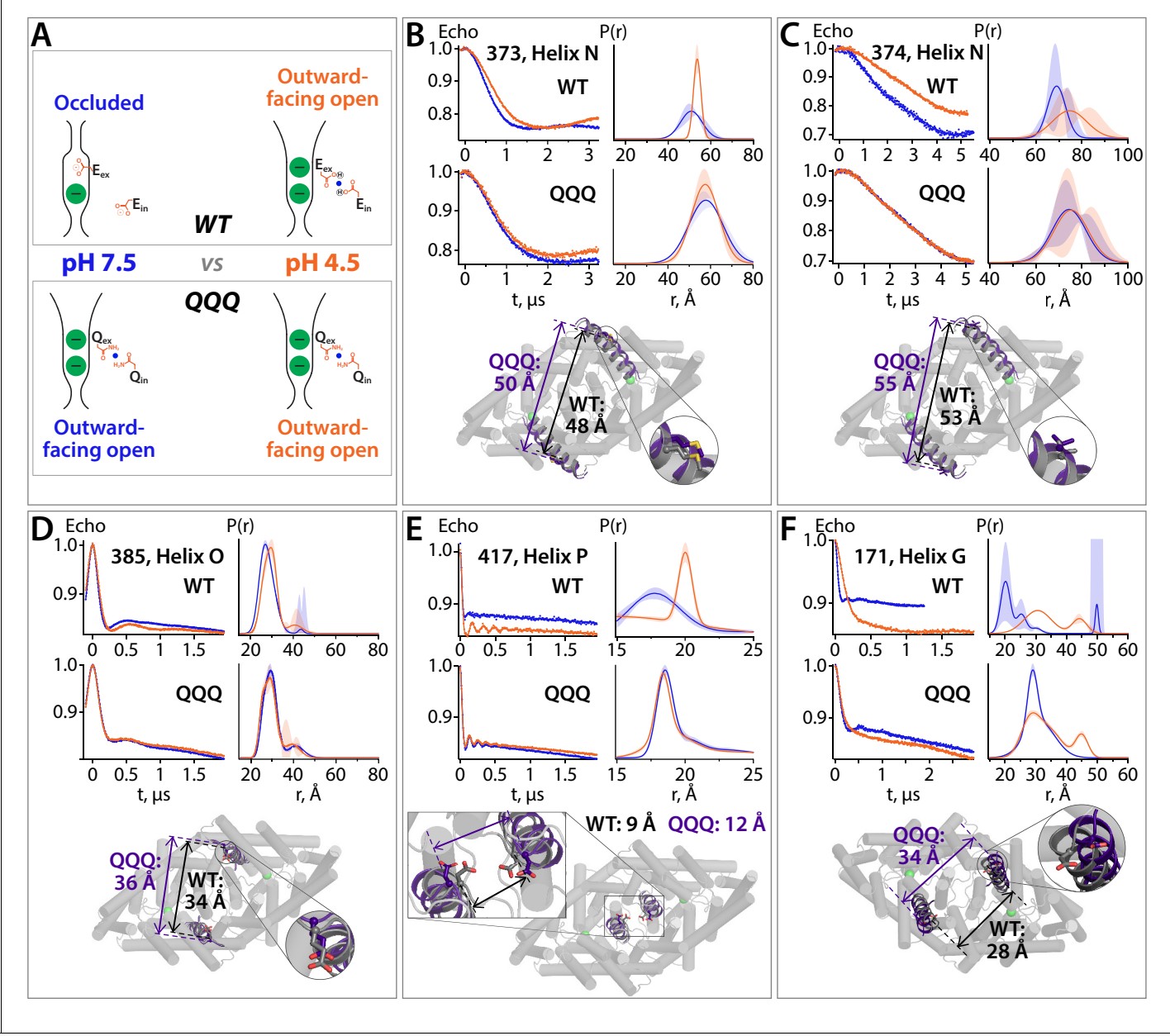

**Figure 7.** DEER spectroscopy reveals that the QQQ conformation is similar to WT CLC-ec1 at low pH. (**A**) Cartoon depiction of the hypothesis. At pH 7.5, the deprotonated Glu$_{ex}$ and Glu$_{in}$ residues will adopt conformations observed in the WT CLC-ec1 crystal structure, while at pH 4.5 they will adopt conformations observed in the QQQ structure, promoting an overall conformational change that leads to widening of the extracellular vestibule. (**B**) – (**F**) DEER measurements on spin-labeled WT and QQQ CLC-ec1 (both in a cysteine-less background). WT CLC-ec1 exhibits pH-dependent changes in inter-subunit distance distributions for spin labels positioned on Helix N, O, P, or G. QQQ CLC-ec1 DEER measurements show little to no pH dependence and have distance distributions similar to those observed with WT at low pH. The lower panels illustrate the position of each labeled residue on the protein and the Cα inter-subunit distances observed in WT versus QQQ. pH-dependent changes at D417C were shown previously for WT (**Khantwal et al., 2016**). Data for samples with spin labels at residue 373, 385, 417, or 171 were acquired using the standard four-pulse protocol; data for the sample labeled at residue 374 were acquired using the five-pulse protocol (**Figure 7—figure supplement 1**).

The online version of this article includes the following source data and figure supplement(s) for figure 7:

**Figure supplement 1.** Five-pulse DEER experiment: breakdown of the fitting procedure.

**Figure supplement 2.** Activity of spin-labeled DEER samples.

**Figure supplement 2—source data 1.** Contains the source data for the absolute values of the transport rates shown normalized in **Figure 7—figure supplement 2**.

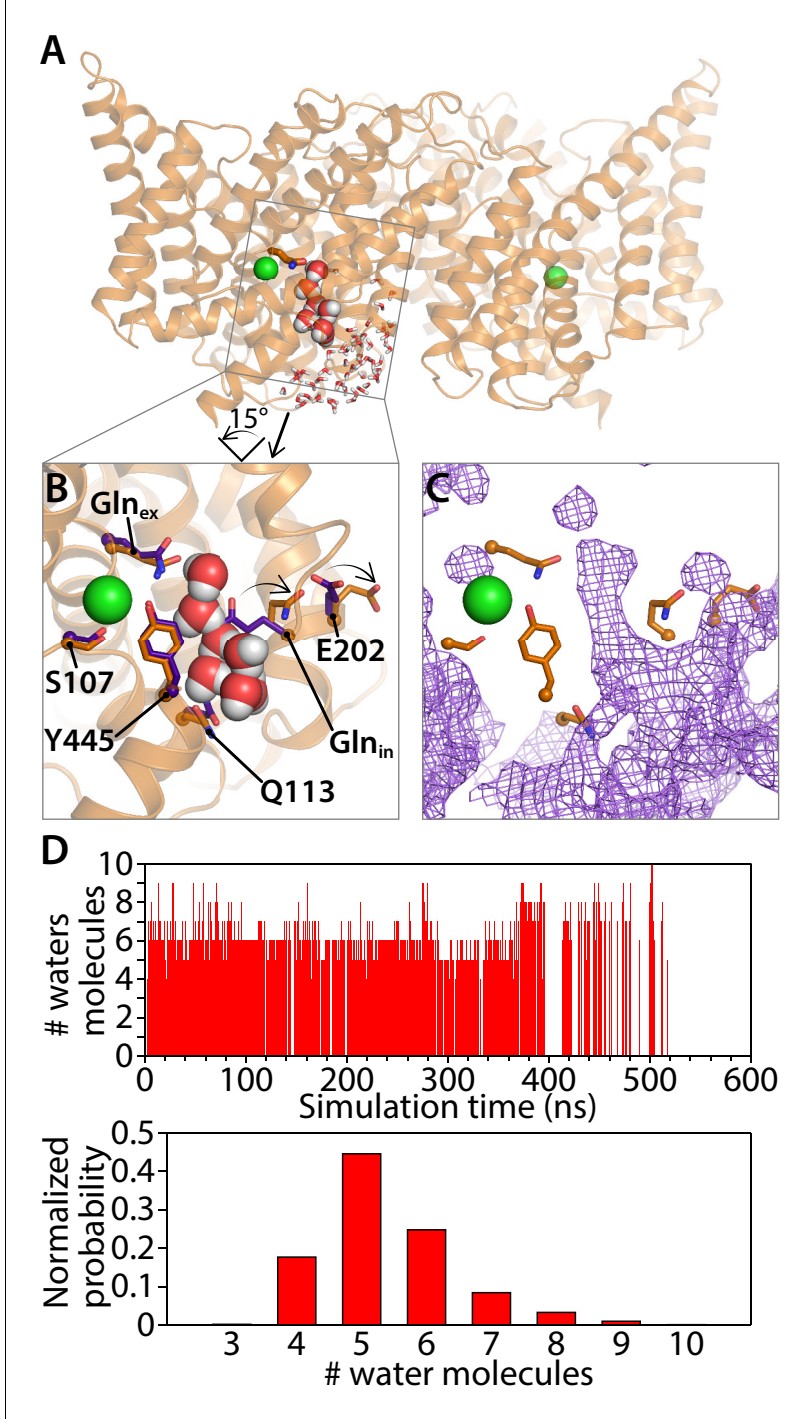

**Figure 8.** Water pathways between Gln$_{ex}$ and intracellular bulk. (**A**) A simulation snapshot showing a continuous water pathway directly connecting Gln$_{ex}$ to the intracellular bulk water in one of the subunits. (**B**) Zoomed in view of the water pathway. In this example, which represents the most frequently observed water pathway (89% of the water pathways), Gln$_{in}$ has rotated away from the position observed in the QQQ crystal structure to make room for the water pathway. The purple side chains show the residue positions observed in the crystal structure; the orange side chains show the residue positions in a representative simulation snapshot. Conserved residue E202 also rotates from its starting position. (**C**) Overall water occupancy map (density contoured at isovalue 0.35) for the side-chain configuration shown in panel B. (**D**) Water pathways between Gln$_{ex}$ and the intracellular bulk water arise spontaneously throughout the 600-ns simulation. Each vertical line in the panels shows the occurrence of a Gln$_{ex}$/bulk-connecting water pathway at that time point, with the length of the line representing the number of water

*Figure 8 continued on next page*

*Figure 8 continued*

molecules in the shortest path. The number of water molecules needed to reach the bulk water follows a normal distribution dominated by 5–6 water molecules.

The online version of this article includes the following source data and figure supplement(s) for figure 8:

**Figure supplement 1.** Water pathways and conformational dynamics of $Gln_{in}$ (Q203) side chain orientation.
**Figure supplement 1—source data 1.** Zip file containing five MD snapshot pdb files representing the Cluster-1 through Cluster-5 water pathways shown in *Figure 8—figure supplement 1*.
**Figure supplement 2.** Residues of interest near the inner water pathway.
**Figure supplement 3.** Conformational dynamics of the E202 side-chain orientation.

over an order of magnitude greater than the occurrence of water wires between $Glu_{in}$ and $Glu_{ex}$ in the WT simulation (1.3%).

The absence of water pathways in the WT simulation is likely due to steric hindrance by $Glu_{in}$, E113, and bulky side chains in the vicinity, which together block direct access of intracellular bulk water toward the protein interior (despite the conformational flexibility of $Glu_{in}$ [*Wang and Voth, 2009*]). In the QQQ simulation, $Gln_{in}$ can equilibrate among five side-chain conformations (Clusters 1–5), all of which can support water pathways (*Figure 8—figure supplement 1A,B*). Most of the water pathways (96% of pathways observed) occur when $Gln_{in}$ is rotated away from its starting conformation (Clusters 1–3), allowing water to flow along a pathway near Q113 (*Figure 8—figure supplement 1C,D*, *Figure 8—figure supplement 1—source data 1*; *Video 3*). In these conformations, the $Gln_{in}$ side chain bends away from Q113 and from the bulky residues F199 and I109, thus allowing intracellular bulk water to enter the protein interior without encountering steric occlusion (*Figure 8—figure supplement 2A*).

The predominant water pathway observed in our simulations is roughly parallel to the Cl⁻ permeation pathway (*Figure 8A*). This pathway for water (and hence H⁺) entry into the protein is different from that previously suggested by us and others. Previously, it was proposed that H⁺ access to the interior of the protein occurs via an entry portal located near the interfacial side of the homodimer (*Lim et al., 2012*; *Han et al., 2014*; *Jiang et al., 2016*) rather than on the 'inner' pathway observed here. While we do see some water pathways occurring along the interfacial route, on a pathway that is lined by $Gln_{in}$, these occur only rarely (*Figure 8—figure supplement 1D*). Importantly, the previous mutagenesis studies supporting the interfacial route are also concordant with the inner water pathway observed here. In the previous studies, mutations that add steric bulk at either E202 (*Lim et al., 2012*) or the adjacent A404 (*Han et al., 2014*) were found to inhibit the H⁺ branch of the CLC-ec1 transport cycle. The observation that all water pathways involve rotation of E202 away from its starting position (*Figure 8B*, *Figure 8—figure supplement 2*, *Figure 8—figure supplement 3*), can explain why bulky mutations at this position would interfere with H⁺ transport.

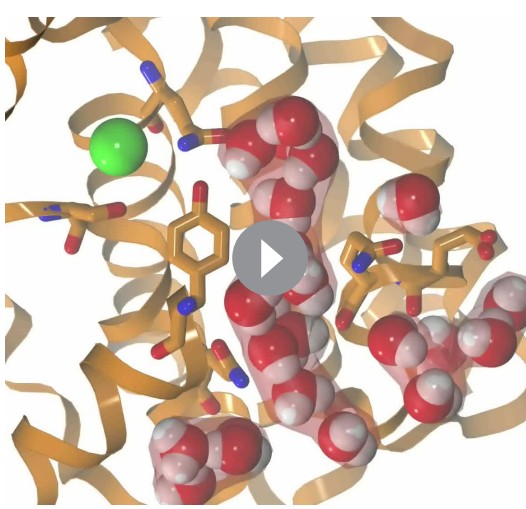

**Video 2.** Dynamics of water pathways and the $Gln_{in}$ side chain in the QQQ mutant simulation. Representative segment of the simulation (250.5–337.5 ns, *Figure 8A*) showing the dramatic hydration of the hydrophobic lumen by water penetration from the intracellular bulk. The $Gln_{ex}$ (Q148) side chain is accessible to the intracellular bulk through the continuous water pathways spontaneously and frequently formed during the simulation. The water pathway conformations undergo dynamical changes in response to the side-chain orientations of $Gln_{in}$. Protein helices are shown as orange ribbons. The Cl⁻ ion at the $S_{cen}$ site is shown as a green sphere. Key amino acids in proximity of the water pathways or involved in Cl⁻-coordination are shown. See also *Figure 8* and Figure 8 supplements.
https://elifesciences.org/articles/53479#video2

## Proton pumping without a titratable residue at the Glu$_{in}$ position

Glu$_{in}$ has long been modeled as a H$^+$-transfer site in the CLC Cl$^-$/H$^+$ mechanism (*Accardi et al., 2005*; *Miller, 2006*; *Lim and Miller, 2009*; *Basilio et al., 2014*; *Accardi, 2015*; *Khantwal et al., 2016*). However, this modeling is contradicted by the observation that several CLC transporter homologs can pump H$^+$ with robust stoichiometry in the absence of a titratable residue at the Glu$_{in}$ position (*Feng et al., 2010*; *Phillips et al., 2012*; *Stockbridge et al., 2012*). In our analysis of water pathways in the MD simulations, we observed that these pathways are not always lined by the Gln$_{in}$ side chain (*Figure 8—figure supplement 1*). This finding strongly suggests that while Glu$_{in}$ *facilitates* water pathways, it is not required as a direct H$^+$-transfer site. To test this hypothesis experimentally, we tested for Cl$^-$-coupled H$^+$ pumping in mutants with non-titratable residues at the Glu$_{in}$ position. We designed the experiment with a large Cl$^-$ gradient and a small H$^+$ gradient, both favoring outward movement of the ions; thus, any H$^+$ transport into the vesicles must occur via Cl$^-$-driven H$^+$ pumping, not leak (*Figure 9A*). With an Ala residue at the Glu$_{in}$ position (E203A), we observed clear H$^+$ pumping above the background signal; by comparison, in line with conventional modeling, the Glu$_{ex}$ mutant E148Q exhibited H$^+$ signals similar to control vesicles (*Figure 9B,C*; *Figure 9—source data 1*). While the coupling stoichiometry of E203A is somewhat degraded compared to the WT protein (*Figure 9D*), the thermodynamic fact arising from this experiment is that H$^+$ pumping occurs with a non-titratable residue at Glu$_{in}$.

In the original pioneering study by Accardi et al., Glu$_{in}$ was identified as a H$^+$-transfer site following a mutagenesis scan of titratable residues at the intracellular side of CLC-ec1 (*Accardi et al., 2005*). Of the 10 mutants tested, only the Glu$_{in}$ mutant E203Q completely abolished detectable H$^+$ transport. To directly compare to this original study, we evaluated E203Q in our assay. As with E203A, we found that E203Q can pump H$^+$; however, its coupling stoichiometry is substantially more degraded (Cl$^-$/H$^+$~75, *Figure 9—figure supplement 1*; *Figure 9—source data 1*). Therefore, it is not too surprising that H$^+$ pumping was not detected in the original study. We also examined double mutants, missing titratable residues at both E203 and its hydrogen-bonding partner E113. When these positions are substituted with two alanines, or with the residues found in the cmCLC homolog (Thr and Lys), H$^+$ pumping is retained (*Figure 9—figure supplement 1*). Thus, the conventional thinking of Glu$_{in}$ as a H$^+$-transport site must be re-evaluated.

## Discussion

In this study, we aspired to determine the high-resolution structure of the CLC 'outward-facing open' conformational state. We approached this problem by using a triple mutant, QQQ, to mimic the conformation in which three key Glu residues are in the protonated state. In principle, it should be possible to obtain a similar structure with WT CLC-ec1 at low pH. Though a crystal structure from a low-pH well-solution was previously reported (*Dutzler et al., 2002*), the pH in the protein crystal may have been closer to that of the purification buffer (pH 7.5) if mixing was inefficient. In an attempt to definitively capture the low-pH state of WT CLC-ec1, we purified the protein at pH 4.0 and performed crystallization trials with well-solutions varying in pH from 3.4 to 5.6. Multiple crystallization attempts, including a grid screen search around reported conditions, varying the protein concentration from 10 mg/mL up to 50 mg/mL, were unsuccessful. In contrast, at pH 7.5–9.5 (the pH range typically used to crystallize CLC-ec1 proteins), we and others routinely

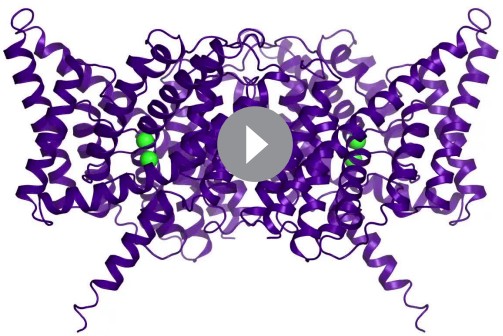

**Video 3.** The inner water pathway. Video shows the starting conformation of QQQ, with residues Gln$_{ex}$, Gln$_{in}$, and E113Q shown as sticks. The same residues in the WT structure are then shown for comparison, along with the side-chain conformational changes that occur in the MD simulation to allow formation of water pathways from the intracellular side. The water pathway shown corresponds to Cluster 1 in *Figure 8—figure supplement 1D*.
https://elifesciences.org/articles/53479#video3

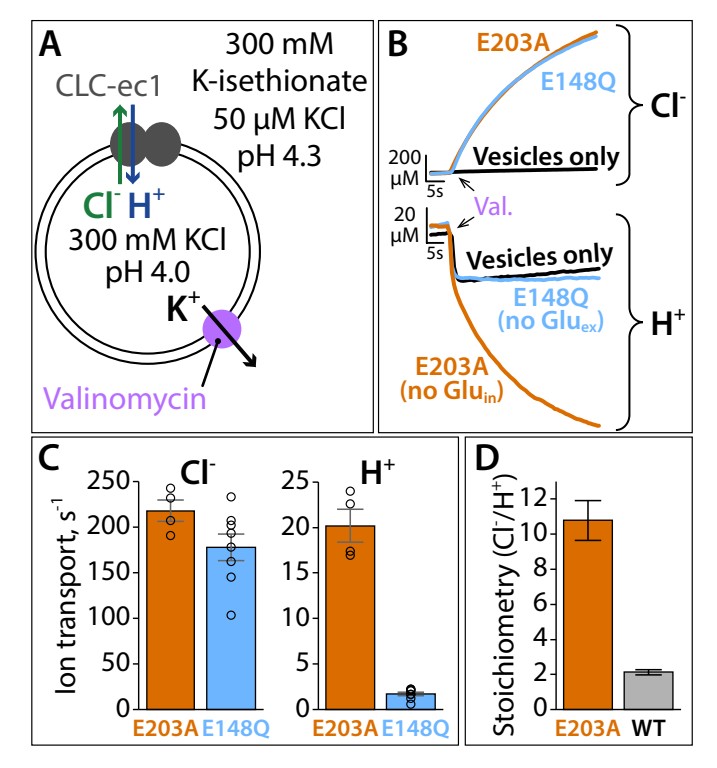

**Figure 9.** Proton pumping by CLC-ec1 with neutralized $Glu_{in}$ (E203A). (**A**) Cartoon depiction of the $H^+/Cl^-$ flux assay. Extravesicular $[Cl^-]$ and $[H^+]$ are simultaneously measured using Ag·AgCl and pH electrodes, respectively. The experimental setup involves a 2-fold gradient for $H^+$, such that any leak will involve movement of $H^+$ out of the vesicles, and any $H^+$ movement into the liposomes must occur via transport coupled to the $Cl^-$ gradient (pumping). (**B**) Representative $Cl^-$ and $H^+$ traces (upper and lower panels, respectively) for CLC-ec1 mutants with a neutral residue at either $Glu_{in}$ (E203A) or $Glu_{ex}$ (E148Q). (**C**) Summary data showing $Cl^-$ and $H^+$ transport rates, average ± SEM. For E203A, n = 4 (samples from two independent protein preparations); for E148Q, n = 8 (samples from four reconstitutions done with protein from three preparations). (**D**) $Cl^-/H^+$ stoichiometry for E203A, with WT CLC-ec1 shown for comparison. Stoichiometry is determined from the ratio of the ion transport rates (panel C). WT samples are from experiments done on the same days as the E203A and mutant samples shown in *Figure 9—figure supplement 1*, with n = 10 (samples from five reconstitutions done with protein from four preparations). The online version of this article includes the following source data and figure supplement(s) for figure 9:

**Source data 1.** Contains the source data for the transport rates shown in *Figure 9* and *Figure 9—figure supplement 1*.
**Figure supplement 1.** Proton pumping by CLC-ec1 variants with substitutions at $Glu_{in}$ and E113.

obtain crystals with 10–20 mg/mL protein (*Dutzler et al., 2002*; *Lim et al., 2012*; *Khantwal et al., 2016*). Given the challenges in crystallizing WT CLC-ec1 at low pH, the QQQ triple mutant presents a viable and appealing alternative approach to capture the desired outward-facing conformation.

Our conclusion that the QQQ mutant structure represents such an intermediate in the CLC transport mechanism is supported by several pieces of evidence. First, the WT protein under low-pH conditions (glutamate residues protonated), adopts a conformation different from the high-pH condition and similar to the conformation adopted by the QQQ mutant, as detected by DEER spectroscopy (*Figure 7*). Second, the movement of Helix P in the QQQ structure (*Figure 7E*) is consistent with the inhibition of $Cl^-/H^+$ transport by Helix P cross-linking (*Khantwal et al., 2016*). Third, the measured transport rates in QQQ relative to E148Q match our expectation for faster $Cl^-$ transport, which was predicted based on the wider extracellular $Cl^-$ entryway in the QQQ structure compared to E148Q (*Figures 2* and *4*). Fourth, the prediction based on MD simulations on the structure – that $Glu_{in}$ residue need not be titratable – was validated experimentally (*Figure 9*). Fifth, and perhaps most compellingly, the details observed in this conformational state reconcile a multitude of findings in the literature, as will be discussed below.

Our results demand that the paradigm of $Glu_{in}$ as a $H^+$-transfer site be re-evaluated. While E203Q has a severely degraded coupling ratio, it retains the ability to pump $H^+$ (*Figure 9—figure supplement 1*); thus, a titratable site at the $Glu_{in}$ position is not an absolute requirement for $H^+$ pumping. Moreover, the more robust coupling observed with E203A (*Figure 9*) is near that observed upon mutation of D278 and K131 (*Accardi et al., 2005*), titratable residues located on the opposite side of the $Cl^-$ permeation pathway relative to E203 ($Glu_{in}$). Therefore, we conclude that the loss of coupling efficiency upon mutation of E203 (and D278 and K131) is not due to loss of a $H^+$-transfer site but rather due to an effect on the efficiency of water-pathway formation or on some other aspect of the transport cycle. This conclusion brings clarity to previous results with $Glu_{in}$ mutants. In a study evaluating transport by 15 mutants with different residues substituted at $Glu_{in}$, it was found that all titratable residues supported well coupled $Cl^-/H^+$ transport, with a pH dependence identical to that of the WT protein even with Lys or His at the $Glu_{in}$ position (*Lim and Miller, 2009*). In that study, slow $H^+$ transport was observed with neutral substitutions at $Glu_{in}$, but it was uncertain at that time whether this transport was a result of *bona fide* $H^+$ pumping or an experimental artifact (*Lim and Miller, 2009*). Interpreted in the light of our current results, this previous study also supports a model in which the $Glu_{in}$ residue contributes indirectly to the efficiency of the $H^+$ transport step.

Our observation of $Glu_{ex}$ in the 'out' position – observed previously only in a channel homolog – was unanticipated. However, in retrospect, it should not have been surprising. Compelling rationale for the close connection between CLC transporter and channel mechanisms was made over a decade ago (*Miller, 2006*), with CLC channels proposed to operate as 'broken' transporters. Moreover, it has been shown that at least some channel homologs retain the ability to transport $H^+$ as part of the gating cycle (*Lísal and Maduke, 2008*). Therefore, it should not be surprising that CLC channels and transporters share a similar conformation for the $Glu_{ex}$ $H^+$-transfer residue. Indeed, the Accardi and Berneche labs recently came to that conclusion based on predictions and tests of their MD simulations (*Leisle et al., 2020*).

The water pathways we identified connect $Gln_{ex}$ directly to the intracellular bulk water. The majority of these pathways (>94%) are approximately parallel to the $Cl^-$-permeation pathway and lined by residue Q113; a small but non-negligible fraction of pathways occurring along an 'interfacial' pathway, lined by residue E202 (*Figure 8—figure supplement 1*). Previously, it was concluded that the interfacial pathway likely predominates, based on the observation that mutation of E113 (or nearby polar residues) to non-polar residues inhibits $H^+$ transport by only 4–7 fold, while mutation of E202 inhibits $H^+$ transport up to 500-fold (*Lim et al., 2012*). These mutagenesis results, however, are also consistent with our MD simulation results, given that E202 conformational dynamics are critical to *both* water pathways (*Figure 8—figure supplement 3*).

## A proposed unifying framework for the CLC transport mechanism

Based on the information gleaned from our study of the QQQ structural intermediate, we propose an updated framework for understanding 2:1 $Cl^-/H^+$ exchange by CLC transporters. This updated model is inspired by four key findings. First, the outward-facing state has improved accessibility for $Cl^-$ to exchange to the extracellular side (*Figure 2*). This state had been previously predicted (*Khantwal et al., 2016*) but is now seen in molecular detail. Second, the protonated $Glu_{ex}$ can adopt an 'out' conformation, within the hydrophobic core of the protein (*Figure 1D,E*). This novel conformation allows us to eliminate a disconcerting step that was part of all previous models: movement of a protonated (neutral) $Glu_{ex}$ – in competition with $Cl^-$ – into the $S_{cen}$ anion-binding site (*Miller and Nguitragool, 2009*; *Feng et al., 2012*; *Basilio et al., 2014*; *Khantwal et al., 2016*). Third, water pathways can connect $Gln_{ex}$ (and presumably $Glu_{ex}$) directly to the intracellular solution (*Figure 8*). Finally, $H^+$ pumping does not require a titratable residue at $Glu_{in}$ (*Figure 9*). Together, these findings allow us to propose a revised framework for the $Cl^-/H^+$ exchange model, which maintains consistency with previous studies and resolves lingering problems.

In our revised model (*Figure 10A*), the first three states are similar to those proposed previously (*Miller and Nguitragool, 2009*; *Feng et al., 2012*; *Basilio et al., 2014*; *Khantwal et al., 2016*). State A reflects the structure seen in WT CLC-ec1, with $Glu_{ex}$ in the 'middle' conformation, occupying $S_{ext}$, and a $Cl^-$ occupying $S_{cen}$. Moving clockwise in the transport cycle, binding of $Cl^-$ from the intracellular side displaces $Glu_{ex}$ by a 'knock-on' mechanism (*Miller and Nguitragool, 2009*), pushing it to the 'up' position and making it available for protonation from the extracellular side (State B). Protonation generates state C, which reflects the structure seen in E148Q CLC-ec1 where $Gln_{ex}$ mimics the protonated $Glu_{ex}$. This sequence of $Cl^-$ binding and protonation is consistent with the experimental finding that $Cl^-$ and $H^+$ can bind simultaneously to the protein (*Picollo et al., 2012*). Subsequently, a protein conformational change generates an 'outward-facing open' state (D). While this state had previously been postulated (*Khantwal et al., 2016*), the QQQ structure presented here provides critical molecular details.

State D involves a widening of the extracellular vestibule, which will facilitate $Cl^-$ binding from and release to the extracellular side. In the QQQ structure (our approximation of State D), the reorientation of Helix N results in subtle changes in $Cl^-$-coordination at the $S_{ext}$ site (*Figure 2—figure supplement 2*), which suggests that binding at this site may be weakened, though we currently lack direct evidence for this conjecture. Regardless of the affinity at $S_{ext}$, the opening of the extracellular permeation pathway in State D will promote $Cl^-$ exchange in both directions, which is essential to achieving reversible transport.

In addition to involving a widening of the extracellular vestibule, state D has the protonated $Glu_{ex}$ in an 'out' conformation and within ~5 Å of $Glu_{in}$ (*Figure 6F*). At first glance, this positioning suggested to us that $Glu_{in}$ might be participating in an almost direct hand-off of $H^+$ to and from $Glu_{ex}$, through an intervening water molecule. However, MD simulations revealed that $Gln_{in}$ is highly

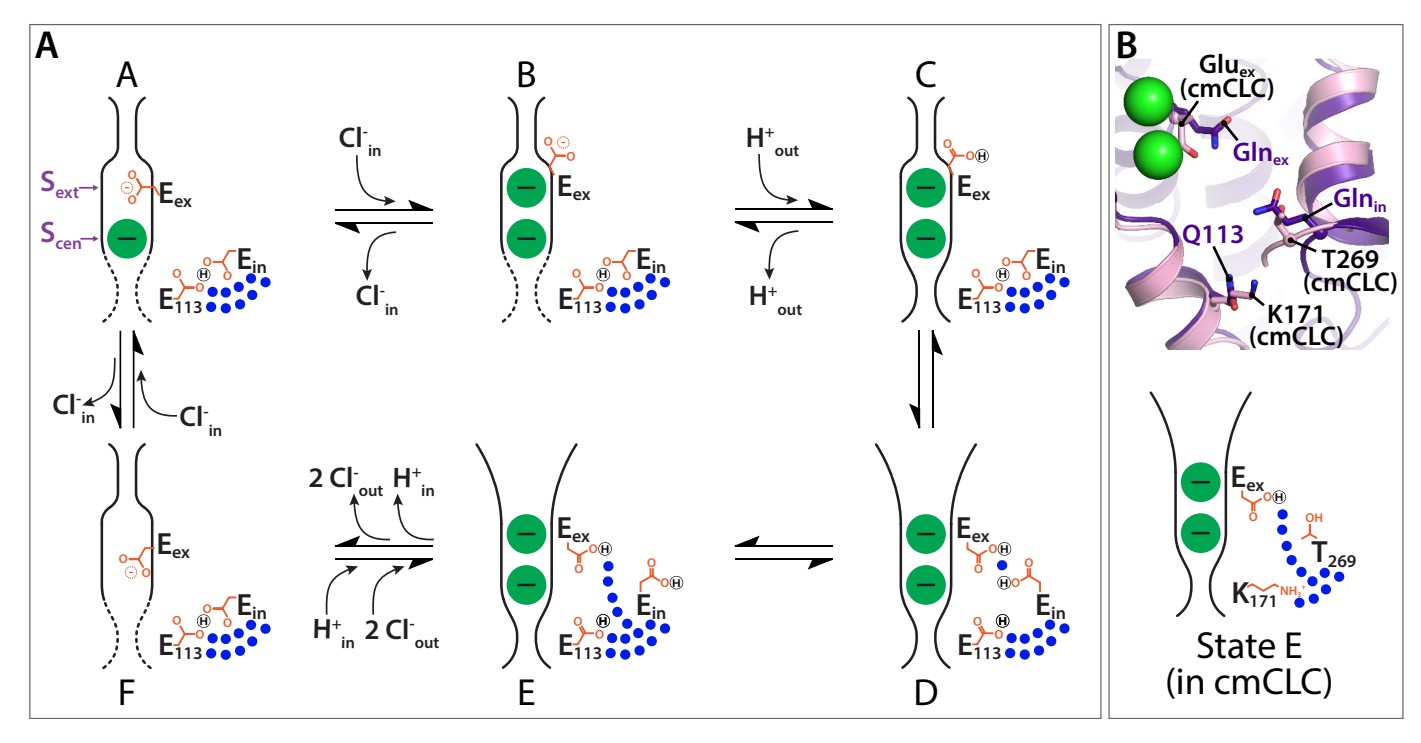

**Figure 10.** Proposed new framework for the CLC Cl⁻/H⁺ transport cycle. (**A**) Starting from state A, which reflects the structure seen in WT CLC-ec1, a single Cl⁻ is bound, and Glu$_{ex}$ is in the 'middle' position. Glu$_{in}$ and E113 are in a H-bonded configuration, restricting water access to the center of the protein. Moving clockwise, binding of Cl⁻ from the intracellular side displaces Glu$_{ex}$ by a 'knock-on' mechanism (*Miller and Nguitragool, 2009*), making it available for protonation from the extracellular side (State B). This protonation step generates state C, which reflects the structure seen in E148Q CLC-ec1, with Glu$_{ex}$ in the 'up' conformation. A subsequent H⁺-induced conformational change generates a state D (captured in the QQQ mutant structure), which has an open extracellular vestibule and new positionings for Glu$_{ex}$ and Glu$_{in}$. Conformational dynamics of Glu$_{in}$ allows water pathways to connect Glu$_{ex}$ directly to the intracellular bulk water, as in state E. From State E, deprotonation of Glu$_{ex}$ promotes its return to the anion pathway, in competition with Cl⁻ (state F). Binding of Cl⁻ from the intracellular side, coordinated with inner-gate opening (*Basilio et al., 2014*) (reflected by the dotted lines) generates the original state A. (**B**) The updated transport model is consistent with Cl⁻/H⁺ exchange seen in transporters that do not have a titratable residue at the Glu$_{in}$ position, such as cmCLC. The upper panel shows a structural overlay of QQQ and cmCLC, highlighting the positions of the Thr and Lys residues occurring at the Glu$_{in}$ and E113 positions respectively. The lower panel is a cartoon depiction of the water pathway.

dynamic and most often is rotated away from its starting position, allowing the robust formation of water pathways from the intracellular bulk water directly to Gln$_{ex}$ (*Figure 8*) (State E). Once such transfer occurs, the deprotonated Glu$_{ex}$ will be disfavored in the hydrophobic core, and it will compete with Cl⁻ for the S$_{cen}$ anion-binding site, generating State F. Although this conformational state has not been observed crystallographically for CLC-ec1, computational studies found that Glu$_{ex}$ favors the S$_{cen}$ position when there are no Cl⁻ ions bound in the pathway (as in State F), (*Picollo et al., 2012*) and that the 'down' position is in general the preferred orientation for Glu$_{ex}$ (*Mayes et al., 2018*). In addition, a recent structure of an Asp$_{ex}$ CLC-ec1 mutant supports that the carboxylate likes to reach down towards S$_{cen}$, in a 'midlow' position, excluding the presence of Cl⁻ at both S$_{cen}$ and S$_{ext}$ (*Park et al., 2019*), as depicted in State F. From this state, binding of Cl⁻ from the intracellular side (coordinated with inner-gate opening [*Basilio et al., 2014*]) knocks Glu$_{ex}$ back up to S$_{ext}$, generating the original state A. This transport cycle is fully reversible, allowing efficient transport in both directions, as is observed experimentally (*Matulef and Maduke, 2005*).

Our proposed updated transport model (*Figure 10A*), in addition to retaining key features based on previous models (*Miller and Nguitragool, 2009*; *Feng et al., 2012*; *Basilio et al., 2014*; *Khantwal et al., 2016*), unifies our picture of both CLC transporter and channel mechanisms. First, it is compatible with transporters that have non-titratable residues at Glu$_{in}$ and E113. Our simulations and experiments (*Figures 8* and *9*) lead to the conclusion that these residues play a key role in

regulating water pathways rather than in direct hand-off of $H^+$. From this perspective, the evolution of non-titratable residues in either (*Feng et al., 2010*) or both (*Phillips et al., 2012*; *Stockbridge et al., 2012*) of these positions is perfectly sensible. In addition, previous mutagenesis experiments on CLC-ec1 and on mammalian transporters, which demonstrate a surprising tolerance for mutations at $Glu_{in}$ (*Zdebik et al., 2008*; *Lim and Miller, 2009*) now make more sense. Strikingly, the structural positioning of T269 in cmCLC, located at the $Glu_{in}$ sequence position, matches the structural positioning of $Gln_{in}$ in the QQQ mutant, such that side-chain dynamics could facilitate comparable water pathways (*Figure 10B*).

The second unifying feature of our model is that it attests to $Glu_{ex}$ movements being conserved amongst every known type of CLC: 2:1 $Cl^-/H^+$ exchangers, 1:1 $F^-/H^+$ exchangers, and uncoupled $Cl^-$ channels. Previously, an 'out' position for $Glu_{ex}$ had been proposed to be essential to the mechanism of $F^-/H^+$ exchangers (*Last et al., 2018*), which allow bacteria to resist fluoride toxicity (*Stockbridge et al., 2012*). However, such a conformation had not been directly observed, and it was postulated that it may be only relevant to the $F^-/H^+$ branch of the CLC family. Structurally, $Glu_{ex}$ in the 'out' position has previously only been observed in a CLC channel. Thus, this conformation is a unifying feature of CLC channels and transporters. Moreover, this conclusion connotes that all CLC proteins act via a 'windmill' mechanism (*Last et al., 2018*), in which the protonated $Glu_{ex}$ favors the core of the protein while the deprotonated $Glu_{ex}$ favors the anion-permeation pathway. Such a mechanism is preferable to previous 'piston'-type mechanisms, with $Glu_{ex}$ moving up and down within the anion-permeation pathway, which required a protonated (neutral) $Glu_{ex}$ to compete with negatively charged $Cl^-$ ions.

Elements of the transport cycle require future experiments to elucidate details. Prominently, the nature of the inward-facing conformational state remains uncertain. In our model, we indicated inward-opening with dotted lines (*Figure 10*, States F, A, B) to reflect this uncertainty. One proposal is that the inner-gate area remains static and transport works via a kinetic barrier to $Cl^-$ movement to and from the intracellular side (*Feng et al., 2010*). Consistent with this proposal, multiscale kinetic modeling revealed that 2:1 $Cl^-/H^+$ exchange can arise from kinetic coupling alone, without the need for large protein conformational change (*Mayes et al., 2018*). An alternative proposal is that CLCs visit a conformationally distinct inward-open state, based on the finding that transport activity is inhibited by cross-links that restrict motion of Helix O, located adjacent to the inner gate (*Basilio et al., 2014*; *Accardi, 2015*). This putative inward-open state appears distinct from the conformational change observed in the QQQ mutant, as the inter-residue distances for the cross-link pairs (399/432 and 399/259) are unchanged in QQQ relative to WT (*Figure 5—figure supplement 3D*). The details of the kinetic-barrier and conformational-change models, and the need for additional experiments on this aspect of transport, have been clearly and comprehensively discussed (*Accardi, 2015*; *Jentsch and Pusch, 2018*).

Simulations of the QQQ conformational state with the glutamine residues reverted to the native, protonatable glutamate side chains will be needed for full understanding of how protonation and deprotonation of these residues affect the conformational dynamics of side chains and water pathways. In our current model, we propose that $Glu_{in}$ needs to be in the protonated (neutral) state to adopt the position that allows water pathways. This proposal appears harmonious with the hydrophobic nature of the protein core explored by the $Gln_{in}$ side chain and the fact that other CLC homologs use neutral residues at this position (*Feng et al., 2010*; *Phillips et al., 2012*; *Stockbridge et al., 2012*). In addition, the proposal is consistent with MD simulations that show the $Glu_{ex}/Glu_{in}$ doubly protonated state is highly populated (*Mayes et al., 2018*) and can favor formation of water pathways under certain conditions (*Ko and Jo, 2010*). Nevertheless, simulations with glutamate side chains in the QQQ conformational state, together with explicit evaluation of $H^+$ transport (*Wang et al., 2018*; *Duster et al., 2019*), are needed to elaborate details of the $H^+$-transfer steps. In addition, multiscale modeling can expand the picture to include multiple pathways that are likely to occur (*Mayes et al., 2018*). Recognizing the importance of elaborating these details, the results reported here represent an essential and pivotal step toward a complete, molecularly detailed description of mechanism in the *sui generis* CLC transporters and channels.

# Materials and methods

## Key resources table

| Reagent type (species) or resource | Designation | Source or reference | Identifiers | Additional information |
|---|---|---|---|---|
| Strain, strain background (*Escherichia coli*) | XL10 Gold | Agilent | 200314 | Chemically competent cells for DNA propagation |
| Recombinant DNA reagent | pASK-CLC-ec1 (plasmid) | PMID:14718478 | | Plasmid containing CLC-ec1 |
| Chemical compound, drug | n-Decyl-β-D-maltoside | Anatrace | D322 | High-purity detergent for protein purification |
| Chemical compound, drug | Lauryl Maltose Neopentyl Glycol (LMNG) | Anatrace | NG310 | High-purity detergent for protein purification |
| Chemical compound, drug | *E. coli* polar lipid extract | Avanti Polar lipids | 100600C | Lipids for protein reconstitution |
| Chemical compound, drug | Isethionic acid | Wako Chemicals | 350–15765 | Acid form of isethionate, used in transport assays |
| Chemical compound, drug | 1-Oxyl-2,2,5,5,-tetramethylpyrroline-3-methyl methanethio-sulfonate | Fisher Scientific | NC9859662 | Spin label for DEER spectroscopy |
| Sequence-based reagent | forward sequencing primer, pASK/CLC-ec1 | IDT | | 5'-CCACTCCCTATCAGTG-3' |
| Sequence-based reagent | forward sequencing primer, CLC-ec1 internal | IDT | | 5'-GGTGTCATTATGTCGA CCATTATGTACCGG-3' |
| Software algorithm | XDS | PMID:20124692 | RRID:SCR_015652 | Data Processing |
| Software algorithm | AIMLESS | PMID:16369096 | RRID:SCR_015747 | Data Processing |
| Software algorithm | PHASER | PMID:19461840 | RRID:SCR_014219 | Structure determination |
| Software algorithm | Coot | PMID:20383002 | RRID:SCR_014222 | Structure refinement |
| Software algorithm | REFMAC | PMID:15299926 | RRID:SCR_014225 | Structure refinement |
| Software algorithm | HOLE | PMID:9195488 | | Structure analysis |
| Software algorithm | Caver | PMID:23093919 | | Structure analysis |
| Software algorithm | Pymol | Pymol | RRID:SCR_000305 | Structure analysis |
| Software algorithm | Adobe Illustrator | Adobe | RRID:SCR_010279 | Figure generation |
| Software algorithm | Microcal Origin | Microcal | RRID:SCR_002815 | ITC measurements |
| Software algorithm | pClamp 9.0 | Molecular Devices | RRID:SCR_011323 | Transport assays |
| Software algorithm | Igor Pro | Wavemetrics | RRID:SCR_000325 | Transport assays, difference distance matrices |
| Software algorithm | Matlab | Mathworks | RRID:SCR_001622 | DEER analysis |

*Continued on next page*

*Continued*

| Reagent type (species) or resource | Designation | Source or reference | Identifiers | Additional information |
|---|---|---|---|---|
| Software algorithm | PROPKA | PMID:26596171 PMID:21269479 | | MD simulations |
| Software algorithm | VMD | PMID:8744570 | RRID:SCR_001820 | MD simulations |
| Software algorithm | DOWSER | PMID:9162944 PMID:25328496 | | MD simulations |
| Software algorithm | CHARMM-GUI Membrane Builder | PMID:25130509 | | MD simulations |
| Software algorithm | NAMD2.12 | PMID:16222654 | | MD simulations |

## Protein preparation and purification

Mutations were inserted in the WT CLC-ec1 protein using Agilent QuikChange Lightning kit and were confirmed by sequencing. Protein purification was carried out as described (*Walden et al., 2007*), with a few changes depending on the type of experiment. For ITC experiments, QQQ or E148Q were purified in buffer A (150 mM Na-isethionate, 10 mM HEPES, 5 mM anagrade decyl maltoside (DM) at pH 7.5). For crystallization experiments, QQQ was extracted with DM. The detergent was gradually exchanged for lauryl maltose neopentyl glycol (LMNG) during the cobalt-affinity chromatography step. The final size-exclusion chromatography step was performed in a buffer containing LMNG. All detergents were purchased from Anatrace (Maumee, OH). For DEER spectroscopy experiments, cysteine mutations were made on a WT or QQQ cysteine-less background (C85A/C302A/C347S) (*Nguitragool and Miller, 2007*). Proteins were purified under reducing conditions and then labeled with the spin label MTSSL (1-Oxyl-2,2,5,5,-tetramethylpyrroline-3-methyl methane-thio-sulfonate) as described (*Khantwal et al., 2016*).

## Crystallography

Purified QQQ protein was concentrated to at least 30 mg/mL. Concentrated protein was mixed with 1.5 parts (w/w) of monolein containing 10% (w/w) cholesterol using the syringe reconstitution method (*Caffrey and Cherezov, 2009*), to generate a lipidic cubic phase mixture. 25 nL droplets of the mixture were dispensed on glass plates and overlaid with 600 nL of precipitant using a Gryphon crystallization robot (Art Robbins Instruments, Sunnyvale, CA). Crystallization trials were performed in 96-well glass sandwich plates incubated at 16°C. The best crystals were obtained using a precipitant solution consisting of 100 mM Tris (pH 8.5), 100 mM sodium malonate, 30% PEG 400% and 2.5% MPD Crystals were harvested after 3–4 weeks of incubation and flash-frozen in liquid nitrogen without further additives. Figures were prepared using PyMOL and Adobe Illustrator.

## Structure determination and refinement

X-ray diffraction data were collected at APS at GM/CA beamline 23ID-D and were processed using XDS (*Kabsch, 2010*) and AIMLESS (*Evans, 2006*) from the CCP4 suite (*Winn et al., 2011*). Owing to radiation damage, a complete dataset was collected by merging data from three different crystals. Phases were obtained using PHASER (*McCoy et al., 2007*) with PDB ID 1ots as a search model. Iterative refinement was performed manually in Coot (*Emsley and Cowtan, 2004*) and REFMAC (*Murshudov et al., 1997*). The final model contained all residues except those of Helix A due to lack of density for this region of the protein. Helix A is observed in different conformations in the monomeric versus dimeric CLC-ec1 structures, and has no impact on function (*Robertson et al., 2010*).

## Reconstitution and flux assays

Flux assay results presented in this paper required a variety of experimental conditions for reconstitutions and flux assays, summarized in *Table 2*. For flux assays comparing activity at pH 7.5 and 4.5, purified CLC-ec1 were first reconstituted at pH 6. The samples were then aliquoted and pH-adjusted using a 9:1 ratio of sample and the adjustment buffer. This step was taken to eliminate variability

**Table 2.** Buffers used for Reconstitution and Flux Assays.

| Buffer R | Buffer F | Ion flux monitored | Protein-lipid ratio (μg/mg) | Protein per assay (μg) |
|---|---|---|---|---|
| Comparing turnover rates at pH 7.5 and 4.5 (*Figure 4*) | | | | |
| 333 mM KCl, 55 mM Na-citrate, 55 mM Na$_2$HPO$_4$, pH 6.0 pH adjustments by adjustment buffers (10x) (after reconstitution): <br> • 0.16 M citric acid, 0.42 M Na$_3$PO$_4$ (for pH 7.5) <br> • 0.43 M citric acid, 0.29 M Na$_3$PO$_4$ (for pH 4.5) | 333 mM K-isethionate, 55 μM KCl, 55 mM Na-citrate, 55 mM Na$_2$HPO$_4$, pH 6.0 pH adjustments by adjustment buffers (10x): <br> • 0.16 M citric acid, 0.42 M Na$_3$PO$_4$ (for pH 7.5) <br> • 0.43 M citric acid, 0.29 M Na$_3$PO$_4$ (for pH 4.5) | Cl$^-$ | 0.4–0.8 | 0.4–0.8 |
| Testing H$^+$ pumping of mutants (*Figure 9*; *Figure 9—figure supplement 1*) | | | | |
| 300 mM KCl, 40 mM Na-citrate, pH 4.0 | 300 mM K-isethionate, 50 μM KCl, 2 mM Na-citrate, pH 4.3 | H$^+$ and Cl$^-$ | 0.4–5.0 | 0.4–10 |
| Testing DEER samples (*Figure 7—figure supplement 1*) | | | | |
| 300 mM KCl, 40 mM Na-citrate, pH 4.5 | 300 mM K-isethionate, 50 μM KCl, 2 mM Na-citrate, pH 4.5 | Cl$^-$ | 0.4 | 0.4–0.5 |

from separate reconstitutions. For experiments testing H$^+$ pumping in mutants, a pH gradient was used to ensure any measured H$^+$ transport was from H$^+$ pumping and not H$^+$ leak.

To measure the rate of H$^+$ and Cl$^-$ transport in flux assays, purified CLC-ec1 proteins were reconstituted into phospholipid vesicles (*Walden et al., 2007*). *E coli* polar lipids (Avanti Polar Lipids, Alabaster, AL) in chloroform were dried under argon in a round-bottomed flask. To ensure complete removal of chloroform, the lipids were subsequently dissolved in pentane and dried under vacuum on a rotator, followed by further drying (5 min) under argon. The lipids were then solubilized at 20 mg/mL in buffer R (*Table 2*) with 35 mM CHAPS on the rotator for 1.5–2 hr. Purified protein (0.4–5 μg per mg lipids) samples or control buffer solution were then added to the prepared lipid-detergent mix and incubated for 10–20 min. Each protein or control reconstitution was divided into 2–4 samples for dialysis to remove detergent, with three buffer changes over 36–60 hr. Following dialysis, each sample was divided into 2–4 for replicate measurements. For experiments shown in *Figures 4* and *9*, the replicate measurements were averaged to obtain a turnover rate value, and each of these averages was counted as one 'n'. For experiments shown in *Figure 7—figure supplement 1* (DEER samples), the replicate measurements from each sample are shown separately.

Reconstituted vesicles were subjected to four freeze-thaw cycles and were then extruded with an Avanti Mini Extruder using a 0.4 μm-filter (GE Healthcare, Chicago, IL) 15 times. For each assay, 60–120 μL of extruded sample were buffer-exchanged through 1.5- to 3.0 mL Sephadex G-50 Fine resin (GE Healthcare, Chicago, IL) columns equilibrated with buffer F (*Table 2*). Exchange was accomplished by spinning the columns at ~1100 g for 90 s using a clinical centrifuge. The collected sample (80–200 μL) was then added to buffer F (500–600 μL) for flux-assay measurement. Extravesicular [Cl$^-$] and [H$^+$] were monitored using a Ag·AgCl electrode and a pH electrode, respectively. The electrodes were calibrated by known additions of KCl (in 20–136 nmol steps) and NaOH (in 10–50 nmol steps). Sustained ion transport by CLC-ec1 was initiated by addition of 1.7–3.4 μg/mL of valinomycin (from 0.5 mg/mL stock solution in ethanol). At the end of each transport experiment, detergent was added to release all Cl$^-$ from the vesicles. This step served as a quality check to confirm that a reasonable yield of vesicles was obtained following the spin-column step. Samples that exhibited a total Cl$^-$ release (sum of Cl$^-$ released by transport and detergent release)>30% than the average were excluded. Using this criterion, 3 out of the 210 assays performed as part of this study were excluded.

## Isothermal titration calorimetry

Titration isotherms were obtained using a VP-ITC microcalorimeter (MicroCal LLC, Northampton, MA) at 25°C. For the experiment, QQQ or E148Q protein samples were purified in buffer A. Titrant used in the experiment was 30 mM KCl in buffer A. The starting concentration of protein was 15–20 μM, in a volume of 1.5 mL. KCl (30 mM) was syringe-titrated into the sample cell in thirty 10 μL injections. The reference data were obtained by titrating buffer A into the protein-containing solution. Data were analyzed using Origin 7.0 software, with fitting using the 'one set of sites' model (keeping

n = 1). The other thermodynamic parameters were obtained accordingly. Isethionate was chosen as the anion of choice for purification of proteins for the ITC experiments since the QQQ mutant shows aggregation upon purification in tartrate-containing solutions, which were previously used for ITC experiments with WT and mutant variants of CLC-ec1 (*Picollo et al., 2009*; *Khantwal et al., 2016*). The mutant is comparatively stable in isethionate and continues to remain stable throughout the ITC experiment. *Figure 4A* shows the gel filtration chromatograms of the mutants before and after the ITC experiments.

## DEER spectroscopy

CW-spectra were collected on a Bruker EMX at 10 mW power with a modulation amplitude of 1.6G. Determination of the spin concentration of the samples were obtained using Bruker's built-in Spin Quantitation method. The spin concentration is divided by the protein concentration to obtain the labeling efficiency. DEER experiments were performed at 83 K on a Bruker 580 pulsed EPR spectrometer at Q-band frequency (33.5 GHz) using either a standard four-pulse protocol (*Jeschke and Polyhach, 2007*) or a five-pulse protocol (*Borbat et al., 2013*). Analysis of the DEER data to determine P(r) distance distributions was carried out in homemade software running in MATLAB (*Brandon et al., 2012*; *Stein et al., 2015*). In the original five-pulse protocol paper the pure five-pulse signal was obtained by subtracting the artefact four-pulse data (*Borbat et al., 2013*). This method requires the ability to discern clearly the extent of the artefact. For the data in this study, we chose to simultaneously fit the four- and five-pulse data with a single Gaussian component in order to improve accuracy of subtracting the four-pulse artefact (*Figure 7—figure supplement 1*). Confidence bands for the distance distributions were determined using the delta method (*Hustedt et al., 2018*). The confidence bands define the 95% confidence interval that the best fit distance distribution will have. In the case of a Gaussian distribution, the shape of the confidence bands can be non-Gaussian.

## Simulation system setup

The structure of the CLC-ec1 QQQ mutant crystallized in this work at 2.6 Å resolution was used as the starting structure for the MD simulation. The 2 $Cl^-$ ions bound at $S_{cen}$ and $S_{int}$ sites in each of the two subunits were preserved for the simulation. In our initial refinement of the QQQ structure, we had modeled water rather than $Cl^-$ at the $S_{ext}$ site, and therefore the simulation was performed without $Cl^-$ at this site. The pKa of each ionizable residue was estimated using PROPKA (*Olsson et al., 2011*; *Rostkowski et al., 2011*), and the protonation states were assigned based on the pKa analysis at pH 4.5. Specifically, E111, E202, E235, and E414 were protonated in the simulation. All other side chains are in their default protonation state. Missing hydrogen atoms were added using PSFGEN in VMD (*Humphrey et al., 1996*). In addition to the crystallographically resolved water molecules, internal water molecules were placed in energetically favorable positions within the protein using DOWSER (*Zhang and Hermans, 1996*; *Morozenko et al., 2014*), including in a bridging position between $Gln_{ex}$ and $Gln_{in}$. This water was not present in our initial structural model but was subsequently added based on experimental density (*Figure 6E,F*). The QQQ protein was embedded in a POPE lipid bilayer using the CHARMM-GUI Membrane Builder (*Wu et al., 2014*). The membrane/protein system was fully solvated with TIP3P water (*Jorgensen et al., 1983*) and buffered in 150 mM NaCl to keep the system neutral. The resulting systems consisting of ~155,000 atoms were contained in a 164 × 127×98 Å$^3$ simulation box.

## Simulation protocols

MD simulation was carried out with NAMD2.12 (*Phillips et al., 2005*) using CHARMM36 force field (*Klauda et al., 2010*; *Huang and MacKerell, 2013*) and a time step of 2 fs. Periodic boundary conditions were used throughout the simulations. To evaluate long-range electrostatic interactions without truncation, the particle mesh Ewald method (*Darden et al., 1993*) was used. A smoothing function was employed for short-range nonbonded van der Waals forces starting at a distance of 10 Å with a cutoff of 12 Å. Bonded interactions and short-range nonbonded interactions were calculated every two fs. Pairs of atoms whose interactions were evaluated were updated every 20 fs. A cutoff (13.5 Å) slightly longer than the nonbonded cutoff was applied to search for interacting atom pairs. Simulation systems were subjected to Langevin dynamics and the Nosé–Hoover Langevin

piston method (*Nosé, 1984*; *Hoover, 1985*) to maintain constant pressure (p=1 atm) and temperature (T = 310 K) (NPT ensemble).

The simulation system was energy-minimized for 10,000 steps, followed by two stages of 1-ns relaxation. Both the protein and the $Cl^-$ ions in the binding sites were positionally restrained (k = 1 kcal·mol$^{-1}$·Å$^{-2}$) in the first 1-ns simulation to allow the membrane to relax. In the second 1-ns simulation, only the protein backbone and the bound $Cl^-$ ions were positionally restrained (k = 1 kcal·mol$^{-1}$·Å$^{-2}$) to allow the protein side chains to relax. Then a 600-ns equilibrium simulation was performed for the system without any restraint applied.

## Analysis of water pathways

The water pathways between Q148 ($Gln_{ex}$) and the intracellular bulk water was searched using a breadth-first algorithm. In Subunit 2 of the homodimer, $Gln_{ex}$ drifted up and away from the 'out' position at the beginning of the simulation (within five ns), and it did not return to the 'out' position during the simulation. In Subunit 1, $Gln_{ex}$ remained near the 'out' position for the first 400 ns; we focused our analysis of water pathways on this subunit. A distance-based criterion of 2.5 Å for the hydrogen bonds, which was found to be useful and inexpensive in computational terms in a previous study (*Matsumoto, 2007*) was used to determine whether water molecules are connected through continuous hydrogen-bonded network. The water pathway with the smallest number of O-H bonds in each frame was considered as the shortest hydrogen-bonded path. The first water molecule in each water pathway is searched using a distance cutoff of 3.5 Å for any water oxygen atoms near the OE1/NE2 atoms of Q148. The water pathway is considered to reach the intracellular bulk once the oxygen atom of the newly found water molecules is at z < −15 Å (membrane center is at z = 0).

## Acknowledgements

We thank Chris Miller and Martin Prieto for comments on the manuscript. We are grateful to Brian Kobilka for use of the crystallization equipment and to K Chris Garcia for use of the MicroCal ITC instrument. This research was funded by NIH GM113195 (MM, ET, and HSM) and P41-GM104601 (ET). TSC was supported by an American Heart Association Fellowship 17POST33670553. This research used resources of the Advanced Photon Source, a US Department of Energy (DOE) Office of Science User Facility operated for the DOE Office of Science by Argonne National Laboratory under contract no. DE-AC02-06CH11357. The SSRL Structural Molecular Biology Program is supported by the DOE Office of Biological and Environmental Research, and by the National Institutes of Health, National Institute of General Medical Sciences (P41GM103393). We also acknowledge computing resources provided by Blue Waters at National Center for Supercomputing Applications (TJ), and Extreme Science and Engineering Discovery Environment (grant MCA06N060 to ET).

## Additional information

### Funding

| Funder | Grant reference number | Author |
| --- | --- | --- |
| National Institutes of Health | GM113195 | Hassane S Mchaourab<br>Emad Tajkhorshid<br>Merritt Maduke |
| American Heart Association | 17POST33670553 | Tanmay S Chavan |
| U.S. Department of Energy | DE-AC02-06CH11357 | Antoine Koehl |
| National Institutes of Health | P41GM103393 | Irimpan I Mathews |
| University of Illinois at Urbana-Champaign | Blue Waters at National Center for Supercomputing Applications | Tao Jiang |
| Extreme Science and Engineering Discovery Environment | MCA06N060 | Emad Tajkhorshid |
| National Institutes of Health | P41-GM104601 | Emad Tajkhorshid |

The funders had no role in study design, data collection and interpretation, or the decision to submit the work for publication.

## Author contributions
Tanmay S Chavan, Ricky C Cheng, Tao Jiang, Irimpan I Mathews, Richard A Stein, Formal analysis, Investigation, Visualization; Antoine Koehl, Formal analysis, Investigation; Hassane S Mchaourab, Emad Tajkhorshid, Merritt Maduke, Conceptualization, Supervision, Funding acquisition

## Author ORCIDs
Ricky C Cheng (iD) https://orcid.org/0000-0002-5667-6945
Emad Tajkhorshid (iD) https://orcid.org/0000-0001-8434-1010
Merritt Maduke (iD) https://orcid.org/0000-0001-7787-306X

## Decision letter and Author response
Decision letter https://doi.org/10.7554/eLife.53479.sa1
Author response https://doi.org/10.7554/eLife.53479.sa2

## Additional files

### Supplementary files
• Transparent reporting form

### Data availability
Diffraction data have been deposited in PDB under accession code 6V2J.

The following dataset was generated:

| Author(s) | Year | Dataset title | Dataset URL | Database and Identifier |
|---|---|---|---|---|
| Mathews II, Chavan TS, Maduke M | 2020 | Crystal structure of ClC-ec1 triple mutant (E113Q, E148Q, E203Q) | http://www.rcsb.org/structure/6V2J | RCSB Protein Data Bank, 6V2J |

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
