## [Decision Letter]

**Acceptance summary:**

The manuscript presents a description and functional assessment of a new structure of the CLC-ec1 chloride/proton exchanger that represents a conformation that mimics the protonated state of three important glutamate residues. A series of complementary experimental and theoretical approaches paint an extended picture of the transport cycle and the new structure fills some of the predicted conformations as well as presents unexpected results. The authors have extensively revised their manuscript and improved the presentation with a better description and interpretation of results as well as a few new experiments. This is a fine contribution that increases our understanding of these remarkable transporters.

**Decision letter after peer review:**

Thank you for submitting your article "Structural characterization of an intermediate reveals a unified mechanism for the CLC Cl^-^/H + transport cycle" for consideration by *eLife*. Your article has been reviewed by three peer reviewers, including Leon D Islas as the Reviewing Editor and Reviewer #1, and the evaluation has been overseen by Richard Aldrich as the Senior Editor.

The reviewers have discussed the reviews with one another and the Reviewing Editor has drafted this decision to help you prepare a revised submission.

In consultation, the reviewers agreed that the new structure represents a significant advance, but they also concluded that at this stage the structure, calculations and experiments do not support a unified mechanism of transport for CLC transporters. The reviewers request that the paper should focus only on characterization of the global structural changes observed and that the conclusion of a unified global transport mechanism be toned down. This also means that the title of the paper should be changed accordingly. Perhaps the paper can be a short communication instead of a full article.

Summary:

CLC chloride/proton exchangers are important physiological mediators of chloride movement. Several structures have been determined ant these, together with functional studies have painted a picture of the transport cycle. However, some parts of the cycle are lacking and he precise role of protonation/deprotonation of key acidic residues in not fully understood.

In this manuscript Chavan et al. report on a new structure of a mutant of the CLC-ec1 transporter. The structural data is also accompanied by spectroscopic (DEER) measurements of conformational changes induced by pH and chloride transport essays.

The authors show that the triple QQQ mutant might represent a new intermediate state, presumably corresponding to protonated external and internal glutamate residues. This structure additionally shows an enlarged entry pathway for extracellular chloride. Interestingly, DEER experiments show that in WT CLC, low pH induces a conformational change compatible with the QQQ structure.

Essential revisions:

While the new structure is novel and interesting, it is unclear that the additional experiments and simulations pin down this newly proposed mechanism. For instance, while the QQQ construct may have increased water access, it does not move protons, and is in fact in a similar conformation to that found in the CLC-1 channel structure. It seems possible that the water access could pertain to uncoupled transport, or channel-like permeation, rather than a means of efficiently transferring the proton to/from Glu_ex_. Furthermore, the significant observation in this structure is the fact that the two ionizable GLUs are now 5.6 Å; away from one another – a point that appears to become irrelevant in the water accessibility mechanism. Therefore, the presentation of this new mechanism needs to be carefully considered. The following outlines some of the specific questions associated with this study:

1) It is unclear that the QQQ structure represents the protonated structure that pertains to the coupled transport in WT CLC-ec1. While experiments were carried out to show that proton pumping does not require ionization at E203, it still may be required for coupled and efficient transport. A previous study (Lim et al.) showed that three non-ionizable E203 mutants – G, S and Q – show low, near-baseline proton uptake that they considered to be inactive. They interpreted this as weak pumping due to "direct, inefficient proton donation from solvent hydronium ion". This seems to also explain the functional results shown here, with weakly coupled 20:1 Cl^-^/H^+^ transport. Furthermore, the MD simulations of the QQQ construct appear to support this, with a mechanism of increased water access allowing for uncoupled proton transfer directly to the external glutamate. Finally, the "out" conformation of Glu_ex_ is also observed in the channel structure CLC-1, which exhibits uncoupled Cl^-^ transport. Therefore, it is not clear whether the observations from uncoupled transport examples pertain to the tightly coupled transport mechanism in the wild-type transporter.

2) The results from the DEER measurements appear to be qualitative. The data from position 374 and 385 do not appear to agree with the expected distances from the crystal structures. Furthermore, the changes observed are much larger than the expected 2 A; changes. It raises the question of whether the protein is much more dynamic at low pH, and whether it is even possible to correlate the change in distributions to the expected changes in distance. It would be useful to analyze other positions, where no change is expected, for comparison.

3) It appears as though there are many changes in the protein structure, as depicted in the backbone RMSD analysis in Figure 3C. This is very interesting, because it has not been observed in any other CLC structure. Do the helices move independently, or does the inverted topology repeat move as rigid bodies? So far, only limited analysis of the structure has been presented. It would be useful to elaborate on this analysis, including adding a figure overlaying the two structures, showing all of the changes in the protein.

4) "within kissing distance" implies the sidechains are near contact. However, a 5.6 Å; measured distance means that these chemical groups are significantly separated. The description is misleading as it suggests that there is the possibility of the residues handing the proton off to one another. It is more likely that there would have to be a water coordinated between (as is observed in the MD simulation), or further conformational change required to bring these residues closer. This should be clarified throughout the manuscript, especially in the cartoons where it shows the two residues in contact.

5) What are the protonated states used in the MD simulations? This is not stated explicitly but is important for future reproducibility of the simulations.

6) Please provide trajectories, or snapshot PDB files of the final equilibrated configurations.

7) This is a high resolution structure for CLC-ec1. Are there crystallographic waters resolved inside the core of the protein? Or along the pathway? Some analysis of this would be useful to the field.

8) Water wires are mentioned throughout the analysis, however, the snapshots and simulation look like the water is disordered. Can you provide analysis on specific water wire geometries?

9) Subsection “A unifying model for the CLC transport mechanism”, third paragraph – you state that the change in the N helix results in changes in ion coordination that are predicted to decrease affinity of this site. However, the ITC measurements argue against this, since the QQQ construct has a higher affinity. Can you clarify this confusion?

10) The fact that the QQQ mutant represents a stable "native" conformation intermediate that can be captured in a X-ray crystal structure and mimics the H^+^-loaded state of the transporter strongly suggests that it would be possible to obtain the structure of the WT (or E148A) at low pH. This would be much more directly relevant for the conclusions of the authors than the results on a triple mutant.

Please add in the manuscript whether the attempt to crystallize the WT or E148A at low pH has been made and why it did not work.

11) A major difference of this structure compared to previous ones is the larger access to the permeation pathway on the extracellular side with removal of all the constrictions present in the WT and the E148A mutant. However, transport rate is only 2 times faster whereas one would expect a much larger functional effect. The data on the Cl^-^ K_d_ for E148A and QQQ are presented in Figure 3—figure supplement 2 where it is also explained that it was not possible to use Br- to distinguish the binding affinities at different sites. The K_d_s for the 2 mutants are very similar.

More importantly, the K_d_ reported for E148A and the QQQ mutants are very similar. How can this be reconciled with a larger permeation pathway in the QQQ mutant but only a doubling of the transport rate? This casts some question marks on the extent to which the QQQ structure is representative of a native conformation intermediate.

The inconsistency between the similar K_d_ and the doubling of the transport rate (in spite of a larger pore) between E148A and the QQQ mutant should be more clearly acknowledged and on this basis the speculative nature of the conclusions is explicitly stated.

12) In a previous study, Accardi et al. (JGP 2004) reported that the E203A mutation abolished proton transport in CLC-ec1. In the present manuscript the authors state that the double mutant E113A/E203A is able to transport protons. It should be clarified in the manuscript whether this difference is a result of the E113A mutation or if the result contrasts with the ones of Accardi et al. In this case the authors should indicate in the Discussion possible reasons for the discrepancy.

13) Is it possible that some of the residues observed to change conformation in the QQQ mutant of CLC-ec1 (for example in Helix P and N) are involved, when mutated, in genetic diseases when expressed in human CLC transporters? Could this be a way to establish the relevance of the conclusions on CLC-ec1 for the entire class of CLC proteins?

---

## [Author Response]

We revised the manuscript to address the concerns, including addition of new data to support the QQQ structural model as well as revisions to the text to clarify what has been learned from our experimental and computational results and what aspects of our proposed model require additional (future) testing. We respectfully assert that our manuscript represents a substantive (even pivotal) contribution to the literature and warrants publication as a regular research article. The bulk of the reviewer criticisms reflect uncertainty as to whether the structure represents an on-pathway conformation. We shared this uncertainty, but we consider that the multiple lines of evidence presented here argue strongly for the mechanistic relevance – indeed the central mechanistic relevance – of the structure. We endeavor to support our view in the detailed responses below.

Essential revisions:1) It is unclear that the QQQ structure represents the protonated structure that pertains to the coupled transport in WT CLC-ec1. While experiments were carried out to show that proton pumping does not require ionization at E203, it still may be required for coupled and efficient transport. A previous study (Lim et al.) showed that three non-ionizable E203 mutants – G, S and Q – show low, near-baseline proton uptake that they considered to be inactive. They interpreted this as weak pumping due to " direct, inefficient proton donation from solvent hydronium ion". This seems to also explain the functional results shown here, with weakly coupled 20:1 Cl^-^/H^+^ transport. Furthermore, the MD simulations of the QQQ construct appear to support this, with a mechanism of increased water access allowing for uncoupled proton transfer directly to the external glutamate. Finally, the "out" conformation of Glu_ex_ is also observed in the channel structure CLC-1, which exhibits uncoupled Cl^-^ transport. Therefore, it is not clear whether the observations from uncoupled transport examples pertain to the tightly coupled transport mechanism in the wild-type transporter.

We have revised the text extensively to clarify the points raised in this paragraph.

Further support for the QQQ mutant representing a WT CLC-ec1 conformation. To provide further support for our interpretation that the QQQ represents an intermediate in the transport cycle, we performed additional DEER/EPR experiments to compare the protein conformation of QQQ with WT. Using spin labels on Helices N, O, P, and G, we shown that the QQQ conformation is similar to that adopted by WT at pH 4.5 (Figure 7).

Coupling does not require ionization at E203. We have clarified that our results are definitive in demonstrating that E203 protonation is not required for coupled transport. Lim and Miller, 2009, indeed previously showed that three non-ionizable E203 mutants exhibited low, near-baseline activity. However, they were very explicit in their waffling regarding the interpretation these results. They pointed out that the small signals *might* be due to pumping (coupling) but that they could just as well be assay artifacts. Had they been convinced that their data demonstrated coupling (even weakened 20:1 coupling), they would have concluded at that time that a titratable group at E203 is *not* required for the fundamental coupling mechanism. We note that the H^+^ turnover rates reported in that research article are not unitary turnover rates, and therefore coupling ratios cannot be derived from those measurements. A major strength of our current work is that – motivated by the QQQ structure – we re-examined this question and demonstrated definitively that proton pumping (coupling) does not require titration of E203. Please also see our response to point 12. We have improved our discussion of these points in the manuscript (subsection “Proton pumping without a titratable residue at the Glu_in_ position”, Discussion, third and fifth paragraphs).

Uncoupled proton transfer.Our observation of water access to the external glutamate from the intracellular side is agnostic as to whether it is part of a coupled or uncoupled H^+^ transport process. It is, however, an observation that fits exceedingly well into a model for coupled H^+^ transport, one that is consistent with all experimental results on coupled transport. We note that experimentally we do not observe uncoupled proton transport in the QQQ mutant, nor has anyone reported uncoupled H^+^ transport in any CLC mutant.

The “out” conformation is observed in the CLC-1 channel structure. We note that Park and MacKinnon’s rationale for concluding that the “out” position may be unique to the channels was based on the observation that in previously known CLC transporters the “out” conformation would generate steric clashes, and not on any argument that an “out” conformation necessarily precludes an antiport mechanism. We have added text to the manuscript to explain this point and the point that CLC channel and transporter mechanisms are highly related, and therefore that it is completely reasonable for the two to share many similar mechanistic steps, including the same “out” conformation for Glu_ex_. We have included this point in the manuscript (subsection “New conformation of Glu_ex_” and Discussion, fourth paragraph).

While it is generally accepted that CLC channels are “broken” (uncoupled) transporters, the mechanistic details of which steps are “broken” in the channel uncoupling are not yet established. One possibility that has been suggested is that the inner and outer Cl^–^ gates are no longer opened and closed in a coupled manner. This possibility is supported by the observation that the structures of CLC channels (CLC-1 and CLC-K) show intracellular gates in open-appearing conformations. In the revised manuscript, we have added text and a figure illustrating the fact that the QQQ mutant retains a constricted intracellular gate (subsection “The intracellular barrier remains constricted” and Figure 3), thus supporting its feasibility as representing an intermediate in a coupled transport cycle.

2) The results from the DEER measurements appear to be qualitative. The data from position 374 and 385 do not appear to agree with the expected distances from the crystal structures. Furthermore, the changes observed are much larger than the expected 2 A; changes. It raises the question of whether the protein is much more dynamic at low pH, and whether it is even possible to correlate the change in distributions to the expected changes in distance. It would be useful to analyze other positions, where no change is expected, for comparison.

The DEER measurements are quantitative, not qualitative, and it is possible to relate changes in the crystal structures to changes in the distance distributions (Kazmier et al. (2014) Proc Natl Acad Sci U S A. 111(41):14752-7. doi: 10.1073/pnas.1410431111). As to the absolute magnitude of the changes in the crystal structures and distance distributions, there is not absolute correlation as there is some uncertainty introduced by the potential rotameric state of the spin-label. We have carried out two different modeling methods, MMM and MDDS (Polyhach et al. (2011) Phys Chem Phys. 13(6):2356-66. doi: 10.1039/c0cp01865a; Islam et al. (2013) J Phys Chem B. 117(17):4740-54. doi: 10.1021/jp311723a), that place spin-labeled rotamers on a fixed protein backbone to derive an expected distance distribution (Author response image 1). While neither method perfectly matches the width of the distance distribution, as expected (Alexander et al. (2013) PLoS One. 8(9):e72851. doi: 10.1371/journal.pone.0072851), they support that the changes seen in the DEER data are reflective of the changes in the crystal structure.

To further support the relevance of the DEER measurements, we evaluated an additional position, on the intracellular side, to complement the previous measurements at sites at the extracellular side. Moreover, we added DEER data for spin-labeled QQQ CLC-ec1 at the 5 positions reported for WT. These new data are shown in Figure 7.

**Author response image 1. respfig1:** Modeling results for expected distance distributions using MMM (left panels) or MDDS (right panels). While neither method perfectly matches the width of the distance distribution, as expected (Alexander et al. (2013) PLoS One. 8(9):e72851), they support that the changes seen in the DEER data are reflective of the changes in the crystal structure.

3) It appears as though there are many changes in the protein structure, as depicted in the backbone RMSD analysis in Figure 3C. This is very interesting, because it has not been observed in any other CLC structure. Do the helices move independently, or does the inverted topology repeat move as rigid bodies? So far, only limited analysis of the structure has been presented. It would be useful to elaborate on this analysis, including adding a figure overlaying the two structures, showing all of the changes in the protein.

Thank you for this suggestion. We have included additional descriptions of the structural changes in the main text (subsection “Overall conformational change in QQQ”) along with new figures (Figure 5C, D; Figure 5—figure supplement 3; Figure 6).

4) "within kissing distance" implies the sidechains are near contact. However, a 5.6 Å; measured distance means that these chemical groups are significantly separated. The description is misleading as it suggests that there is the possibility of the residues handing the proton off to one another. It is more likely that there would have to be a water coordinated between (as is observed in the MD simulation), or further conformational change required to bring these residues closer. This should be clarified throughout the manuscript, especially in the cartoons where it shows the two residues in contact.

We have revised the cartoons and removed the “within kissing distance” phrase. To make the narrative easier to follow, we have separated the introduction and discussion of the changes at Glu_ex_ and Glu_in_. Glu_ex_ changes are introduced and discussed in Figure 1 and following; Glu_in_ changes are introduced and discussed in Figure 6and following.

5) What are the protonated states used in the MD simulations? This is not stated explicitly, but is important for future reproducibility of the simulations.

The simulation was performed using the QQQ mutant structure as the template, with three glutamates mutated to glutamine (E148Q, E203Q, E113Q), and therefore there is no concern regarding their protonation states. Regarding the other protonatable residues: since the triple mutant represents the CLC-ec1 structure under low pH, we assigned the protonation state of each titratable residue in the protein based on the pKa analysis at pH 4.5 using PROPKA. To be specific, the following amino acids were in their protonated states in the simulation: E111, E202, E235, and E414. All other side chains are in their default protonation state. We have added this information to the Materials and methods subsection “Reconstitution and chloride flux assay”, first paragraph.

6) Please provide trajectories, or snapshot PDB files of the final equilibrated configurations.

Representative snapshot PDB files are provided in Figure 8—source data 1. The 5 snapshot PDBs in this zip file represent the 5 different water pathways observed in our simulations (Figure 8—figure supplement 1).

7) This is a high resolution structure for CLC-ec1. Are there crystallographic waters resolved inside the core of the protein? Or along the pathway? Some analysis of this would be useful to the field.

Good point. We have added this information in Figure 6—figure supplement 1.

8) Water wires are mentioned throughout the analysis, however, the snapshots and simulation look like the water is disordered. Can you provide analysis on specific water wire geometries?

The water molecules are not disordered, although from the figure it may appear so. In fact, we have characterized these water wires after careful analysis of the water arrangements ensuring that the water molecules are connected and form a H-bonded configuration required for proton transport. We have included the definition of water pathways in “Analysis of water pathways” in “Materials and methods”. The pathways are defined only when there are water molecules connected through a continuous hydrogen-bonded network. The reason that in the figure the water pathway might appear a bit disordered may be due to inclusion of all water molecules in the protein cavity, rather than only the ones that form a single file. We opted to include all water molecules (rather than just showing those forming a single file pathway) to highlight how much the protein interior was hydrated during the simulation. The pdb files we have provided (see response to point #6) allow the reader to examine the pathways closely.

9) Subsection “A unifying model for the CLC transport mechanism”, third paragraph – you state that the change in the N helix results in changes in ion coordination that are predicted to decrease affinity of this site. However, the ITC measurements argue against this, since the QQQ construct has a higher affinity. Can you clarify this confusion?

It is correct that, based on the changes observed at Helix N, we anticipated that the affinity of the QQQ construct for Cl^–^ would be decreased compared to E148Q. There are several very reasonable explanations for the observation that QQQ and 148Q have similar affinities measured by ITC. First, the ITC measurement does not distinguish between Cl^–^ binding to the central site (S_cen_), which appears structurally identical in QQQ and E148Q, and the external site (S_ext_), which exhibits some structural differences. Second, the crystal structures are static snapshots of the proteins, whereas the ITC measurements are performed on protein in detergent solution. Conformational flexibility in solution may allow the S_ext_ binding site in QQQ to make small (~0.5 Å) adjustments that generate an increased binding affinity. We have completely rewritten this section and the section on QQQ and E148Q transport assays (subsection “Comparison of QQQ and E148Q Cl^–^ binding and transport rates”) to clarify the point that even though these mutants display identical Cl^–^ binding affinities (a thermodynamic measurement), they display different Cl^–^ transport rates (a kinetic measurement) and that our results are completely consistent with our interpretation that the Helix-N opening occurs as part of the normal transport cycle. Please also see our response to point #11.

10) The fact that the QQQ mutant represents a stable "native" conformation intermediate that can be captured in a X-ray crystal structure and mimics the H^+^-loaded state of the transporter strongly suggests that it would be possible to obtain the structure of the WT (or E148A) at low pH. This would be much more directly relevant for the conclusions of the authors than the results on a triple mutant. Please add in the manuscript whether the attempt to crystallize the WT or E148A at low pH has been made and why it did not work.

We now describe our crystallization attempted on WT CLC-ec1 at low pH, in the first paragraph of the Discussion. The protein is very soluble at low pH, and we did not obtain any crystals even at 50 mg/mL protein. (At pH 8.5, CLC-ec1 crystallizes from solutions at 10-20 mg/mL.)

11) A major difference of this structure compared to previous ones is the larger access to the permeation pathway on the extracellular side with removal of all the constrictions present in the WT and the E148A mutant. However, transport rate is only 2 times faster whereas one would expect a much larger functional effect. The data on the Cl^-^ K_d_ for E148A and QQQ are presented in Figure 3—figure supplement 2 where it is also explained that it was not possible to use Br- to distinguish the binding affinities at different sites. The K_d_s for the 2 mutants are very similar.More importantly, the K_d_ reported for E148A and the QQQ mutants are very similar. How can this be reconciled with a larger permeation pathway in the QQQ mutant but only a doubling of the transport rate? This casts some question marks on the extent to which the QQQ structure is representative of a native conformation intermediate. The inconsistency between the similar K_d_ and the doubling of the transport rate (in spite of a larger pore) between E148A and the QQQ mutant should be more clearly acknowledged and, on this basis, the speculative nature of the conclusions is explicitly stated.

We thank the reviewers for raising these points, which guided us in making clarifications and improvements in the manuscript, rewriting the section describing these results (subsection “Comparison of QQQ and E148Q Cl^–^ binding and transport rates” and Figure 4). A critical point we had not made clear is that the prediction that Cl^–^ transport through QQQ will be faster than through E148Q is agnostic towards the magnitude of the difference. The wider vestibule in QQQ predicts that transport through QQQ will be faster than through E148Q, if extracellular gate-opening is a rate-limiting step. On the other hand, if Cl^–^ transport through E148Q and QQQ have the same rate-limiting step, then Cl^–^ transport rates should be similar. Experimental measurements revealed that QQQ Cl^–^ transport rates are ~2-fold faster than E148Q transport rates at pH 7.5, the pH at which the binding studies were performed (Figure 4B, C), indicating that the two mutants have different rate-limiting steps. Thus, the prediction is supported. A further prediction is that if this difference in transport rates is due to the wider extracellular Cl^–^ entryway in QQQ compared to E148Q, then lowering the pH (to allow E148Q to adopt the QQQ-like (outward-facing open) conformation) should increase the Cl^–^ transport rate. Consistent with this prediction, transport rates of the E148Q mutant increase by 2-fold at pH 4.5 (Figure 4D). While the transport rates of QQQ and E148Q remain slower than WT CLC-ec1 by ~6-fold, such a result is not surprising (actually, it is expected) given that these Glu_ex_ mutants lack a negatively charged carboxylate to compete Cl^–^ out of the permeation pathway. Thus, these results support the conclusion that E148Q CLC-ec1 (and by extension WT CLC-ec1) undergoes an opening of the extracellular vestibule at low pH. To provide further support that the QQQ structure represents a native conformation intermediate, we additionally added new DEER data comparing QQQ to WT CLC-ec1 (Figure 7).

12) In a previous study, Accardi et al. (JGP 2004) reported that the E203A mutation abolished proton transport in CLC-ec1. In the present manuscript the authors state that the double mutant E113A/E203A is able to transport protons. It should be clarified in the manuscript whether this difference is a result of the E113A mutation or if the result contrasts with the ones of Accardi et al. In this case the authors should indicate in the Discussion possible reasons for the discrepancy.

Thank you for raising the point that we should provide additional information discussion concerning these previous results. We have added new data to the manuscript, reporting flux assay results on single mutants at the Glu_in_ position, E203A and E203Q (Figure 9 and Figure 9—figure supplement 1). We find that both mutants can pump protons. Interestingly, E203A retains stronger coupling stoichiometry than E203Q (10:1 versus 75:1). In the Accardi et al. manuscript (JGP 2005), it is the latter mutant (E203Q) that was examined. There is no discrepancy between our results and those of Accardi et al., whose report of reversal potentials for E203Q (Figure 2C in Accardi et al., 2005) are consistent with a 75:1 coupling stoichiometry. Accardi et al. did not examine the E203A mutant. We have included a discussion of these points in the manuscript, subsection “Proton pumping without a titratable residue at the Glu_in_ position”, last paragraph.

13) Is it possible that some of the residues observed to change conformation in the QQQ mutant of CLC-ec1 (for example in Helix P and N) are involved, when mutated, in genetic diseases when expressed in human CLC transporters? Could this be a way to establish the relevance of the conclusions on CLC-ec1 for the entire class of CLC proteins?

Thank you for this suggestion. We have added relevant references to disease-causing mutations (subsections “Opening of the extracellular vestibule” and “Overall conformational change in QQQ”). We have modified the title to “A CLC-ec1 mutant reveals global conformational change and suggests a unifying mechanism for the CLC Cl^–^/H^+^ transport cycle”.